# RUFY3 and RUFY4 are ARL8 effectors that promote coupling of endolysosomes to dynein-dynactin

Tal Keren-Kaplan [1], Amra Sarić [1], Saikat Ghosh [1], Chad D. Williamson[1], Rui Jia[1], Yan Li [2] & Juan S. Bonifacino [1✉]

The small GTPase ARL8 associates with endolysosomes, leading to the recruitment of several effectors that couple endolysosomes to kinesins for anterograde transport along micro-tubules, and to tethering factors for eventual fusion with other organelles. Herein we report the identification of the RUN- and FYVE-domain-containing proteins RUFY3 and RUFY4 as ARL8 effectors that promote coupling of endolysosomes to dynein-dynactin for retrograde transport along microtubules. Using various methodologies, we find that RUFY3 and RUFY4 interact with both GTP-bound ARL8 and dynein-dynactin. In addition, we show that RUFY3 and RUFY4 promote concentration of endolysosomes in the juxtanuclear area of non-neuronal cells, and drive redistribution of endolysosomes from the axon to the soma in hippocampal neurons. The function of RUFY3 in retrograde transport contributes to the juxtanuclear redistribution of endolysosomes upon cytosol alkalinization. These studies thus identify RUFY3 and RUFY4 as ARL8-dependent, dynein-dynactin adaptors or regulators, and highlight the role of ARL8 in the control of both anterograde and retrograde endolysosome transport.

[1] Neurosciences and Cellular and Structural Biology Division, Eunice Kennedy Shriver National Institute of Child Health and Human Development, National Institutes of Health, Bethesda, MD, USA. [2] Proteomics Core Facility, National Institute of Neurological Disorders and Stroke, National Institutes of Health, Bethesda, MD, USA. ✉email: juan.bonifacino@nih.gov

The ADP-ribosylation factor (ARF) family of small GTPases comprises ~30 members that regulate various aspects of cell physiology[1] (for a complete list of abbreviations, see Supplementary Table 1). Among these members, the mammalian ARL8A and ARL8B paralogs (herein referred to as ARL8 unless otherwise specified) are unique in their ability to associate with endolysosomes and to regulate multiple endolysosomal functions[2] (throughout this article, we use the term endolysosomes broadly to denote various types of lysosomes, late endosomes, and related endolysosomal organelles). Like other small GTPases, ARL8 cycles between GDP-bound, inactive, and GTP-bound, active forms[1,2]. Whereas the GDP-bound form is cytosolic, the GTP-bound form associates with endolysosomes[3,4]. The association of ARL8 with endolysosomes depends on an N-terminal acetylated, amphipathic α-helix[4], which, by analogy with other members of the ARF family[5,6], likely swings out from the rest of the molecule upon GTP binding, to mediate interaction with the lipid bilayer. In addition, this association requires the endolysosome-associated hetero-octameric complex BORC[7], which may function as a guanine-nucleotide-exchange factor (GEF) for the conversion of GDP-bound to GTP-bound ARL8[8].

The regulation of cellular functions by small GTPases is generally mediated by effectors that interact with the GTP-bound forms of the GTPases, resulting in the recruitment and/or allosteric activation of the effectors[1]. Several effectors have been identified for mammalian ARL8, including the hetero-hexameric tethering complex HOPS[9], the adaptor proteins PLEKHM1[10] and PLEKHM2 (also known as SKIP, the name used here)[11,12], and the kinesin-3 motor protein KIF1[13]. The interaction of ARL8-GTP with HOPS promotes fusion of lysosomes with late endosomes[10,14], phagosomes[9], autophagosomes[15], and Salmonella-containing vacuoles[16], in some cases in cooperation with PLEKHM1[10]. The interaction of ARL8-GTP with SKIP mediates recruitment of the kinesin-1 motor protein (a $KIF5_2$-$KLC_2$ hetero-tetramer) for anterograde transport of endolysosomes toward the peripheral cytoplasm in non-polarized cells[7,11,17–19] and toward the distal axon in neurons[20–22]. The ARL8-SKIP-kinesin-1 ensemble is also responsible for the formation of tubular endolysosomes in lipopolysaccharide-treated macrophages[23] and for the process of phagolysosome resolution[24]. Finally, while kinesin-1 requires SKIP for interaction with ARL8-GTP, kinesin-3 interacts directly with ARL8-GTP[13], also promoting anterograde transport of endolysosomes toward the cell periphery in non-polarized mammalian cells[17], as well as endolysosomes, synaptic vesicle precursors, presynaptic active zone proteins, and dense core vesicles in C. elegans and Drosophila neurons[13,21,22,25,26]. Through these interactions, ARL8 regulates various cellular processes mediated by endolysosomes[27], including endocytic degradation[14,28], autophagy[7,10,20,29], microbial killing, and antigen presentation[9], natural killer cell cytotoxicity[19], mTOR signaling[29,30], cell adhesion and migration[31], invasive cancer growth[32], axonal growth-cone dynamics[20], axon branching[33], and egress of β-coronaviruses from infected cells[34].

Although the number of known ARL8 effectors may seem already large, there are ARF-family GTPases that have many more effectors. For example, ARF1 has more than 15 known effectors[1,35]. It is thus possible that ARL8 has a larger set of effectors and functions than are currently known. Herein we report the results of a search for additional ARL8 effectors using MitoID[36], a method involving proximity biotinylation with mitochondrially targeted forms of human ARL8A and ARL8B, followed by isolation of biotinylated proteins and their identification by mass spectrometry. Using this method, we identified the <u>RU</u>N and <u>FYVE</u> domain-containing protein RUFY3[37] as an ARL8 effector. We also found that RUFY3 and the related RUFY4 protein promote ARL8-dependent distribution of endolysosomes to the juxtanuclear area of non-polarized cells, and from the axon to the soma of rat hippocampal neurons. Further biochemical and

cellular analyses demonstrated that RUFY3 and RUFY4 also interact with the retrograde microtubule motor dynein-dynactin. These findings thus revealed that both RUFY3 and RUFY4 are ARL8 effectors that promote the coupling of endolysosomes to dynein-dynactin for retrograde transport along microtubules. ARL8 can thus regulate both anterograde and retrograde endolysosome transport through interactions with kinesin and dynein-dynactin motors, respectively.

## Results

**Identification of RUFY3 and RUFY4 as ARL8 effectors**. To identify ARL8 effectors, we used a modification of the MitoID proximity biotinylation method previously developed to identify interactors of RAB GTPases[36] (Supplementary Fig. 1a). This method involves targeting of bait proteins to mitochondria, providing for a more uniform background of non-specific hits[36]. Bait constructs were made by attaching a mitochondrial-targeting sequence (MTS) from the outer mitochondrial membrane protein TOM20[38], followed by the BioID2 biotin ligase[39], to the N-terminus of the GTP-bound, active (Q75L) or GDP-bound, inactive (T34N) forms of human ARL8A and ARL8B[3] lacking the N-terminal amphipathic α-helix (Mito-ARL8 constructs) (Fig. 1a). MTS-BioID2 without ARL8 was used as a negative control (Fig. 1a). The Mito-ARL8 and control constructs were expressed by transient transfection into HEK293T cells, after which cells were incubated with 50 μM biotin for 24 h. Cells were then extracted with detergent, and biotinylated proteins captured on NeutrAvidin-agarose beads and identified by mass spectrometry (Supplementary Fig. 1a). Three biological replicates were used per sample. Data were analyzed by comparing the abundance of proteins labeled by MTS-BioID2-ARL8-Q75L relative to the MTS-BioID2 control vs. MTS-BioID2-ARL8-T34N relative to the MTS-BioID2 control for both ARL8A (Fig. 1b) and ARL8B (Fig. 1c) (see also Supplementary Data 1). A top hit in these analyses was the known ARL8 effector PLEKHM2 (SKIP)[11], which was only detected in isolates from the Q75L forms of both ARL8A (Fig. 1b) and ARL8B (Fig. 1c). The identification of SKIP verified the reliability of the assay.

Another top hit for both the ARL8A (Fig. 1b) and ARL8B (Fig. 1c) constructs was a protein named RUFY3 (also known as SINGAR1, RIPX or ZFYVE30)[40] (Fig. 1d). RUFY3 is one of four members of the RUFY family of proteins in humans[37]. These proteins comprise (in N- to C-terminal direction) RUN, coiled-coil (CC) and FYVE domains joined by disordered sequences (Fig. 1d). RUFY3, in particular, has two CC domains (CC1 and CC2) and exists as 6 spliceforms, two of which are the 620-amino-acid RUFY3.1 (transcript variant 1) (NM_001037442) (also known as RUFY3XL[37]) and the 469-amino-acid RUFY3.2 (transcript variant 2) (NM_014961) (Fig. 1d). Whereas RUFY3.1 includes all the domains of the RUFY family, RUFY3.2 lacks a C-terminal region comprising part of the CC2 domain and the entire FYVE domain (Fig. 1d). The shorter RUFY3.2 is the only RUFY3 spliceform characterized to date, with previous studies showing that it plays roles in neuronal polarity and axon formation/degeneration[40–43], and in cancer cell migration, invasion, and metastasis[44–47]. Protein and mRNA expression databases (https://www.proteinatlas.org/search/rufy3) indicate that RUFY3 is expressed in all cells and tissues, although with higher expression in the brain. Our mass spectrometric analyses identified five unique peptides derived from the longer RUFY3.1 spliceform (Supplementary Fig. 1b), demonstrating that this particular species is expressed in HEK293T cells.

To confirm the identification of RUFY3 as an ARL8 effector and to determine whether other members of the RUFY family also interact with ARL8, we examined the intracellular localization of

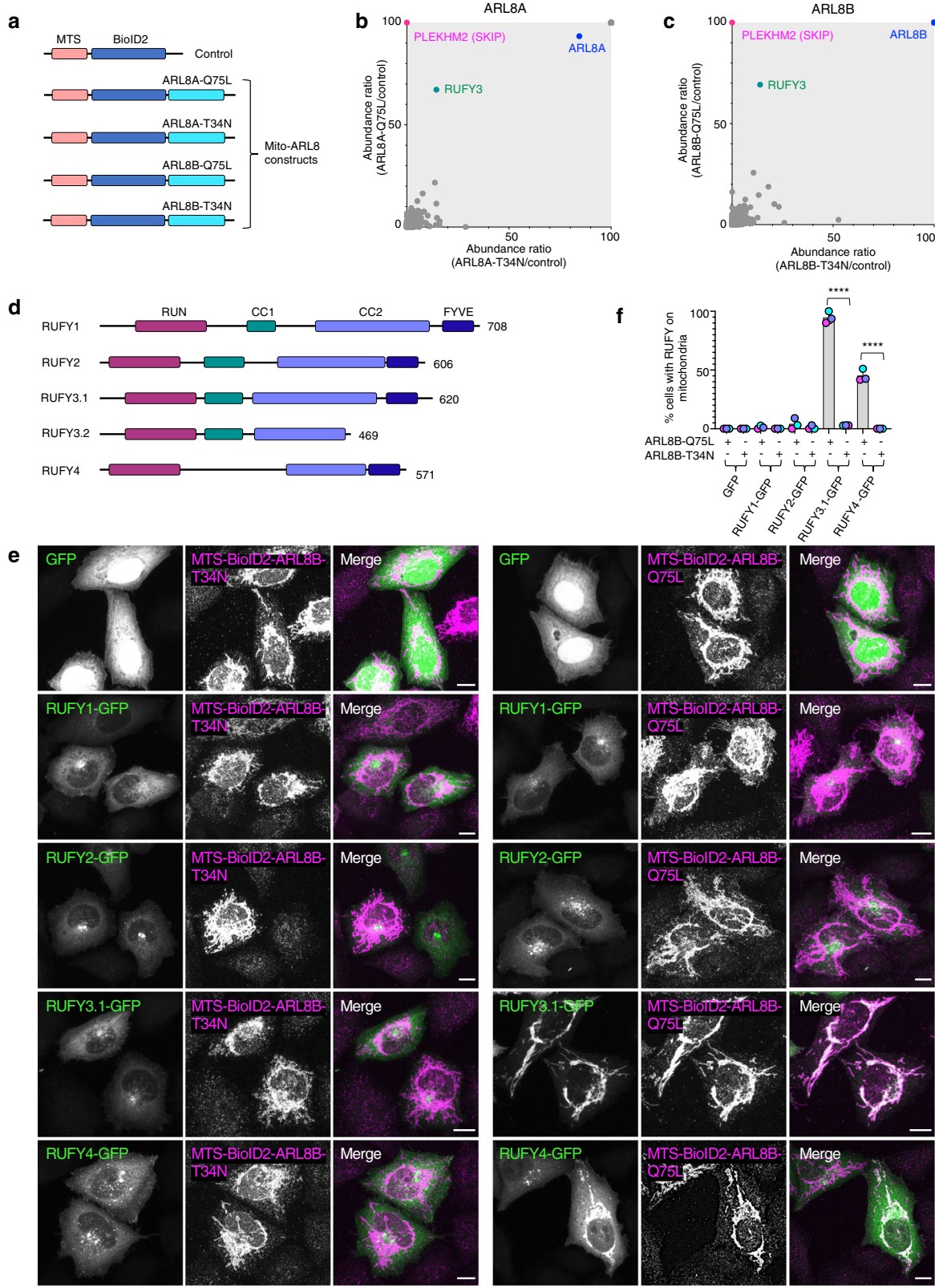

GFP-tagged forms of the RUFY proteins co-expressed with MTS-BioID2-ARL8B-T34N and MTS-BioID2-ARL8B-Q75L in HeLa cells (Fig. 1e, f). We observed that, in the presence of the GDP-bound MTS-BioID2-ARL8B-T34N, GFP-tagged RUFY1, RUFY2, RUFY3.1 and RUFY4 localized to a cluster of vesicles in the juxtanuclear area of the cell (Fig. 1e, f). However, in the presence of the GTP-bound MTS-BioID2-ARL8B-Q75L, GFP-tagged RUFY3.1 and RUFY4 redistributed to mitochondria, whereas GFP-tagged RUFY1 and RUFY2 retained their juxtanuclear distribution (Fig. 1e, f). In contrast to RUFY3.1-GFP, RUFY3.2-GFP was completely cytosolic under all conditions (Supplementary Fig. 1c, d), probably because it lacks part of the CC2 domain and the FYVE domain

**Fig. 1 Identification of RUFY3 and RUFY4 as ARL8 effectors. a** Schematic representation of control and Mito-ARL8 constructs used in MitoID. Mito-ARL8 constructs comprise the mitochondrial targeting sequence (MTS) from TOM20[38], followed by the BioID2 biotin ligase[39], and ARL8A and ARL8B lacking the N-terminal α-helix and harboring the activating Q75L or inactivating T34N mutations. Data from 3 biological replicates were used for the assay. **b** Graph showing the relative abundance of hits identified by mass spectrometry for MTS-BioID2-ARL8A-Q75L/MTS-BioID2 control vs. MTS-BioID2-ARL8A-T34N/MTS-BioID2 control using MitoID. **c** Same as (**b**) for MTS-BioID2-ARL8B-Q75L/MTS-BioID2 control vs. MTS-BioID2-ARL8B-T34N/MTS-BioID2 control. Hits of interest in panels (**b**) and (**c**) are highlighted. **d** Domain organization of RUFY proteins in N- to C-terminal direction. RUN: RPIP8, UNC-14 and NESCA domain, CC1: coiled-coil 1 domain, CC2: coiled-coil 2 domain, FYVE: Fab1, YOTB, Vac1 and EEA1 domain. Amino-acid numbers are indicated. RUFY3.1 and RUFY3.2 are two spliceforms of RUFY3. **e** Immunofluorescence microscopy of HeLa cells co-expressing GFP or RUFY-GFP fusion proteins (green) with MTS-BioID2-ALR8B-Q75L or MTS-BioID2-ALR8B-T34N. Fixed cells were stained with antibody to BioID2 (magenta) and imaged by confocal microscopy. Single channels are shown in grayscale. Scale bars: 10 μm. **f** Quantification of the percentage of cells in which RUFY proteins were re-localized to mitochondria in experiments such as that in panel (**e**). Values are the mean ± SD from three independent experiments (minimum of 300 cells per condition). Statistical significance was calculated using one-way ANOVA with multiple comparisons between groups using Tukey's test. ****
$p < 0.0001$. See also Supplementary Fig. 1 and Supplementary Data 1.

(Fig. 1d). We also found that mitochondrially targeted GDP-bound and GTP-bound forms of another endolysosomal small GTPase, RAB7A (RAB7A-T22N-BirA*-HA-MAO and RAB7A-Q67L-BirA*-HA-MAO, respectively)[36], failed to re-localize GFP-tagged RUFY3.1 and RUFY4, whereas they did re-localize the known RAB7A effector RILP tagged with GFP[48], to mitochondria (Supplementary Fig. 1e, f). These observations further demonstrated the specificity of the interactions of RUFY3.1 and RUFY4 with GTP-bound ARL8.

To further corroborate these interactions, we performed pull-down assays using recombinant GST-ARL8B proteins and FLAG-tagged RUFY proteins expressed by transient transfection in HEK293T cells (Fig. 2a). We observed that GST-ARL8B-Q75L, but not GST-ARL8B-T34N, pulled down both RUFY3.1-FLAG and RUFY4-FLAG (Fig. 2a). In contrast, neither GST-ARL8B protein pulled down RUFY1-FLAG and RUFY2-FLAG (Fig. 2a). Furthermore, we examined the co-immunoprecipitation of FLAG-tagged RUFY proteins with endogenous ARL8A and ARL8B in HEK293T cells (Fig. 2b). FLAG-tagged forms of the dynein-dynactin adaptor HOOK1[49] and the ARL8 effector SKIP[11] were used as negative and positive controls, respectively. These experiments showed that RUFY3.1-FLAG and RUFY4-FLAG specifically co-immunoprecipitated endogenous ARL8A and ARL8B, whereas RUFY1-FLAG and RUFY2-FLAG did not (Fig. 2b). In agreement with the mitochondrial re-localization experiments (Supplementary Fig. 1c, d), the shorter RUFY3.2-showed little or no pulldown with GST-ARL8B-Q75L (Supplementary Fig. 1g, h) and co-immunoprecipitation with ARL8A and ARL8B (Fig. 2b).

Taken together, these experiments demonstrated that both RUFY3.1 and RUFY4 have the ability to interact with GTP-bound, but not GDP-bound, ARL8, suggesting that they are bona fide ARL8 effectors. RUFY1 and RUFY2, on the other hand, did not bind to either form of ARL8, ruling out their function as ARL8 effectors. Because RUFY3.2 was cytosolic (Supplementary Fig. 1c, d) and did not interact with ARL8 (Fig. 2b; Supplementary Fig. 1g, h), this spliceform was not used in further experiments, and RUFY3.1 was simply referred to as RUFY3.

RUFY4 (also known as ZFYVE31) is expressed at low levels in most tissues and cells, with the exception of the brain, lung, and lymphatic organs (https://www.proteinatlas.org/search/rufy4), probably explaining why it was not identified in the MitoID experiments using HEK293T cells. Nevertheless, because it behaves as an ARL8 effector, we performed some experiments with this protein as well.

**The CC2 domain of RUFY3 mediates binding to ARL8.** We next sought to identify the region of RUFY3 that mediates interaction with ARL8. To this end, we generated deletion constructs of RUFY3-GFP (Fig. 2c) and co-expressed them with

MTS-BioID2-ARL8B-Q75L in HeLa cells (Fig. 2d). Interaction with ARL8 was inferred from re-localization of the RUFY3 constructs to mitochondria. By analogy with SKIP, which inter-acts with ARL8 via the RUN domain[11,12,18], we expected the homologous RUN domain of RUFY3 to be important for this interaction. However, we found that deletion of the RUN, CC1 or FYVE domains had no effect on the re-localization of RUFY3-GFP to mitochondria (Fig. 2d, e). Likewise, combined deletion of the RUN and CC1 domains did not prevent the re-localization of RUFY3-GFP to mitochondria (Fig. 2d, e). In contrast, deletion of the CC2 domain, alone or in combination with the FYVE domain, abrogated the re-localization of RUFY3-GFP to mitochondria (Fig. 2d, e). Moreover, we observed that the CC2 domain alone, but not the FYVE domain alone, was sufficient for re-localization to mitochondria (Fig. 2d, e).

In line with the mitochondrial re-localization experiments, we found that recombinant GST-ARL8B-Q75L pulled down RUFY3-FLAG constructs lacking the RUN, CC1, or FYVE domain expressed in HEK293T cells (Fig. 2f, g). In contrast, GST-ARL8B-Q75L did not pull down RUFY3-FLAG constructs lacking the CC2 domain or the CC2-FYVE tandem (Fig. 2f, g). Furthermore, the CC2 domain alone, but not the RUN or FYVE domains alone, was sufficient for pull down by GST-ARL8B-Q75L (Fig. 2f-i). GST-ARL8B-T34N did not pull down any of the constructs, confirming that the interactions involving the CC2 domain are specific for the GTP-bound form of ARL8B.

From these experiments, we concluded that the interaction of RUFY3 with GTP-ARL8B is mediated by the CC2 domain.

**Requirement of ARL8 for association of RUFY3/4 with a juxtanuclear vesicle cluster.** We next examined whether RUFY3 and RUFY4 co-localize with normal, non-mitochondrially targeted ARL8B. Because none of the antibodies that we tested detect the endogenous proteins by immunofluorescence microscopy, we examined the localization of fluorescently-tagged versions of the proteins expressed by transient transfection in HeLa cells (Fig. 3a, b). In cells with low-to-moderate expression levels, we observed that both RUFY3-GFP and RUFY4-GFP largely co-localized with ARL8B-mCherry on a cluster of vesicles adjacent to the nucleus, as well as some scattered vesicles (Fig. 3a-c).

Double knock out (KO) of ARL8A and ARL8B (ARL8A-B KO)[18] partially reduced the association of RUFY3-GFP with vesicles, and the labeled vesicles appeared more dispersed (Fig. 3d; Supplementary Fig. 2a). Co-transfection of the ARL8A-B-KO cells with a plasmid encoding ARL8B-Q75L-mCherry restored full association of RUFY3-GFP with juxtanuclearly clustered vesicles (Fig. 3d). Therefore, ARL8 is partially required for the recruitment of RUFY3 to vesicles, and, in addition, promotes the juxtanuclear clustering of the vesicles.

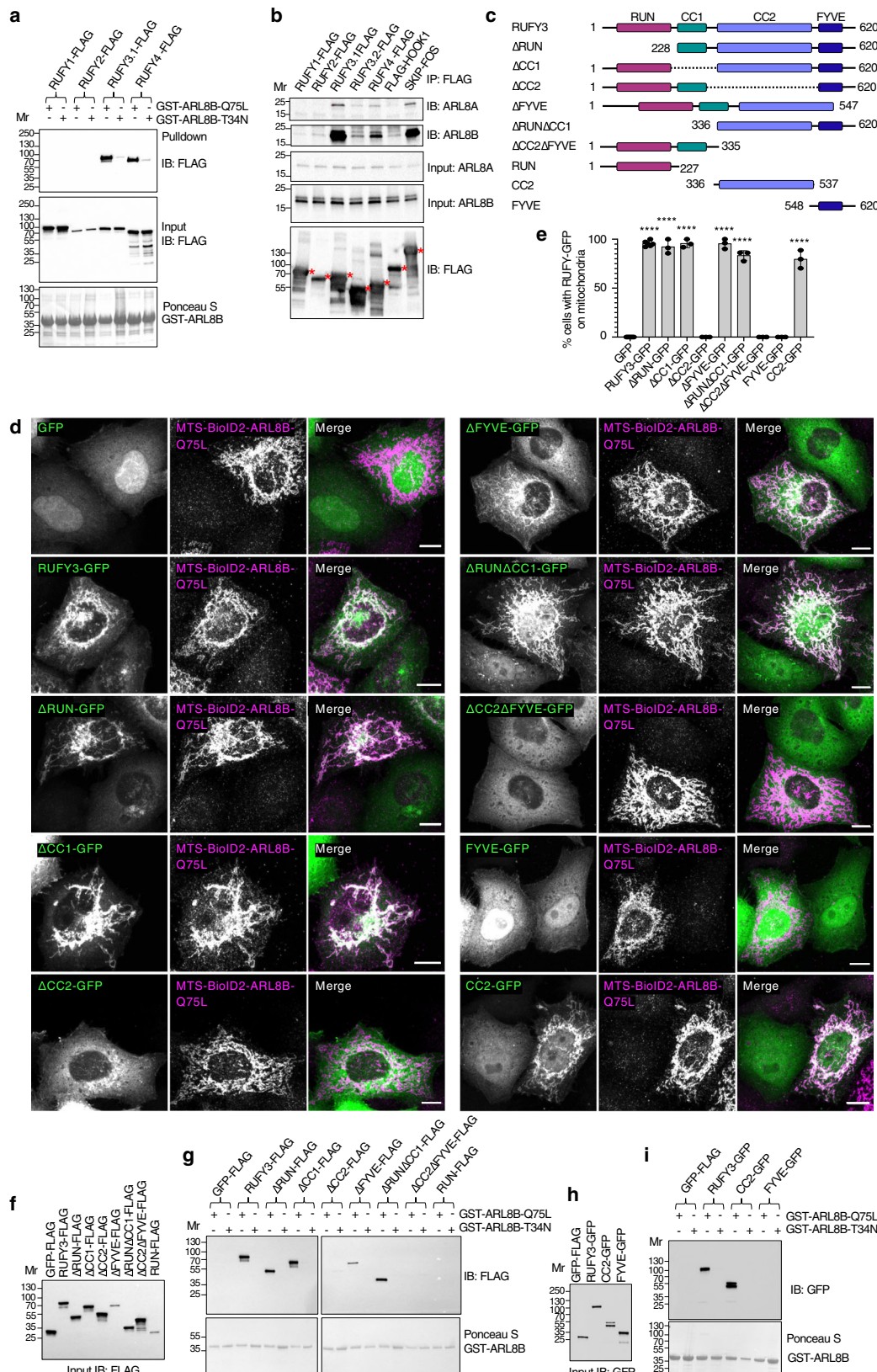

RUFY4-GFP was almost completely dissociated from vesicles upon ARL8A-B KO, save for a few juxtanuclear puncta (Fig. 3e; Supplementary Fig. 2b). Co-transfection with the plasmid encoding ARL8B-Q75L-mCherry rescued association of RUFY4-GFP with the vesicles (Fig. 3e). Thus, RUFY4 appears even more dependent on ARL8 for association with vesicles.

**Contribution of the CC2 and FYVE domains for association of RUFY3 with vesicles.** To further dissect the requirement of different RUFY3 domains for recruitment to vesicles, we examined by live-cell imaging the intracellular localization of GFP-tagged versions of the RUFY3 deletion mutants depicted in Fig. 2c. We observed that RUFY3-GFP constructs lacking the RUN and/or

**Fig. 2 Biochemical evidence for binding of RUFY3 and RUFY4 to ARL8, and dissection of RUFY3 domains required for ARL8 binding. a** GST-ARL8B-Q75L and GST-ARL8B-T34N were used to pull down the indicated RUFY-FLAG proteins expressed by transfection in HEK293T cells. FLAG-tagged proteins were identified by immunoblotting (IB) and GST proteins by Ponceau S staining. **b** Extracts of HEK293T cells transfected with plasmids encoding the indicated FLAG- or FOS (FLAG-one-strep)-tagged proteins were immunoprecipitated (IP) with anti-FLAG, and immunoblotted (IB) for endogenous ARL8A and ARL8B and the FLAG tag. Red asterisks indicate the positions of the different FLAG- or FOS-tagged proteins (expected molecular masses: RUFY1-FLAG, 80.8 kDa; RUFY2-FLAG, 71 kDa; RUFY3.1-FLAG, 71.1 kDa; RUFY3.2-FLAG, 54 kDa; RUFY4-FLAG, 65 kDa; FLAG-HOOK1, 86 kDa; SKIP-FOS, 116 kDa). Data in (**a**) and (**b**) are representative of 2 experiments with similar results. **c** Schematic representation of RUFY3 deletion constructs. Domain organization is as depicted in Fig. 1b. Amino-acid numbers are indicated. Δ stands for deletion. Constructs were tagged with GFP or the FLAG epitope. **d** Immunofluorescence microscopy of HeLa cells expressing GFP or the RUFY3-GFP deletion constructs shown in panel **c** (green) together with MTS-BioID2-ALR8B-Q75L. Cells were fixed and stained as described in Fig. 1e. Scale bars: 10 μm. **e** Quantification of the percentage of cells in which RUFY-GFP proteins were re-localized to mitochondria from experiments such as that shown in panel (**d**). Values are the mean ± SD from a minimum of three independent experiments, each scoring a minimum of 300 cells per condition. Statistical significance compared to cells expressing GFP was calculated using one-way ANOVA with multiple comparisons with Dunnett's test. ****$p < 0.0001$. **f, g** GST-ARL8B-Q75L and GST-ARL8B-T34N were used to pull down the indicated RUFY3-FLAG deletion constructs expressed by transfection in HEK293T cells. FLAG-tagged proteins (**f, g**) were detected by immunoblotting (IB) for the FLAG epitope. Input GST-ARL8B-Q75L and GST-ARL8B-T34N were detected by Ponceau-S staining. **h, i** GST-ARL8B-Q75L and GST-ARL8B-T34N were used to pull down the indicated RUFY3-GFP deletion constructs expressed by transfection in HEK293T cells. GFP-tagged proteins (**h, i**) were detected by immunoblotting (IB) for GFP. Input GST-ARL8B-Q75L and GST-ARL8B-T34N (**i**) were detected by Ponceau-S staining. The experiments in (**f–i**) are representative of 2 experiments with similar results. Mr represents molecular mass (kDa).

CC1 domains (ΔRUN, ΔCC1 and ΔRUNΔCC1) were largely associated with the juxtanuclear cluster (Fig. 3f–h). RUFY3-GFP constructs lacking the CC2 domain (ΔCC2) were less associated with vesicles, and these vesicles were more dispersed in the cytoplasm (Fig. 3f–h). This phenotype was similar to that of full-length and ΔCC2 RUFY3-GFP constructs expressed in ARL8A-B-KO cells (Fig. 3d, Supplementary Fig. 2a), consistent with the CC2-ARL8 interaction promoting both membrane recruitment and juxtanuclear clustering of RUFY3-GFP–decorated vesicles. Deletion of the FYVE domain resulted in a protein that was largely cytosolic, except for a tight perinuclear punctum (Fig. 3f–h). This residual punctum completely disappeared when the ΔFYVE-GFP construct was expressed in ARL8A-B-KO cells (Supplementary Fig. 2a), or when a ΔCC2ΔFYVE-GFP construct was expressed in WT cells (Fig. 3f, g), indicating that it reflected association of the CC2 domain with ARL8. Finally, we observed that GFP fusions to the CC2 or FYVE domains alone were cytosolic, demonstrating that they are insufficient for association with vesicles (Fig. 3f, g). From these observations, we concluded that both the CC2-ARL8 interaction and the FYVE domain contribute to the association of RUFY3 with vesicles, and that the CC2-ARL8 interaction additionally promotes juxtanuclear clustering of the vesicles. Since the isolated CC2 or FYVE domains are cytosolic, however, other domains may also contribute to the membrane recruitment and function of RUFY3. The ability of the CC2 domain alone to be recruited to mitochondria by the MTS-BioID2-ARL8B-Q75L construct (Fig. 2d, e) can be likely explained by the overexpression of this construct, a situation that is different from the expression of the CC2 domain in cells having endogenous levels of ARL8 (Fig. 3f, g).

**RUFY3 and RUFY4 promote juxtanuclear clustering of endolysosomes.** Because ARL8 was previously shown to associate with endolysosomes[3,4], we examined if the vesicles containing associated RUFY3-GFP and RUFY4-GFP also contained the endogenous endolysosomal membrane protein LAMP1 in HeLa cells. Indeed, in cells expressing low-to-moderate levels of RUFY3-GFP and RUFY4-GFP, we observed partial but significant co-localization of these proteins with LAMP1 (Fig. 4a, b). Moreover, we noticed that cells overexpressing RUFY3-GFP and RUFY4-GFP exhibited more juxtanuclear clustering of endolysosomes relative to cells overexpressing only GFP (Fig. 4c, d). For RUFY4, these observations agree with a previous report by Terawaki et al.[50].

Expression of the different RUFY3-GFP deletion mutants (Fig. 2c) showed that those that bound ARL8 (i.e., constructs lacking the RUN, CC1 or FYVE domains) caused juxtanuclear clustering, whereas those that did not bind ARL8 (i.e., constructs lacking the CC2 domain) failed to cause juxtanuclear clustering of endolysosomes (Fig. 5a, b), thus demonstrating a correlation between ARL8 binding and juxtanuclear clustering of endolysosomes by RUFY3.

Conversely, knock down (KD) of RUFY3 mRNA in HeLa cells using a pool of four siRNAs (Fig. 5c) caused dispersal of LAMP1 toward the cell periphery (Fig. 5d–f), including cell vertices (Fig. 5d, arrows). Treatment with each of the individual siRNAs in the pool showed that three of the four siRNAs were effective at knocking down RUFY3 (Supplementary Fig. 3a), and those same three caused peripheral redistribution of LAMP1 (Supplementary Fig. 3b–d), including localization to cell vertices (Supplementary Fig. 3b, arrowheads), confirming the specificity of the result using pooled siRNAs.

RUFY4 mRNA could not be detected by qRT-PCR of HeLa cells (Fig. 5c), consistent with the low expression levels of this mRNA in most cell lines (https://www.proteinatlas.org/ENSG00000188282-RUFY4/celltype). For this reason, the effect of RUFY4 KD in these cells was not tested.

From these experiments, we concluded that RUFY3 and RUFY4 promote localization of endolysosomes to the juxtanuclear area of the cell.

**RUFY3 or RUFY4 overexpression promotes retrograde transport and decreases the abundance of axonal endolysosomes.** To examine the effect of RUFY3 and RUFY4 on endolysosome movement, we turned to rat hippocampal neurons, in which axonal microtubules are uniformly arrayed, with their minus ends in the soma and their plus ends in the distal axon[51]. We observed that, in fixed neurons, both RUFY3-FLAG and RUFY4-FLAG partially co-localized with ARL8B-mCherry on vesicles containing the endogenous endolysosomal marker LAMTOR4[52] and transgenic LAMP1-GFP in both the axon and dendrites (Fig. 6a–d).

Live-cell imaging and kymograph analysis of axonal vesicle movement showed that, in neurons co-expressing GFP (control), LAMP1-RFP–containing vesicles moved in both anterograde (from the proximal to the distal axon) and retrograde (from the distal to the proximal axon) directions in approximately equal proportions (Fig. 6e, f). Co-expression with RUFY3-GFP or RUFY4-GFP reduced the total number of LAMP1-RFP tracks

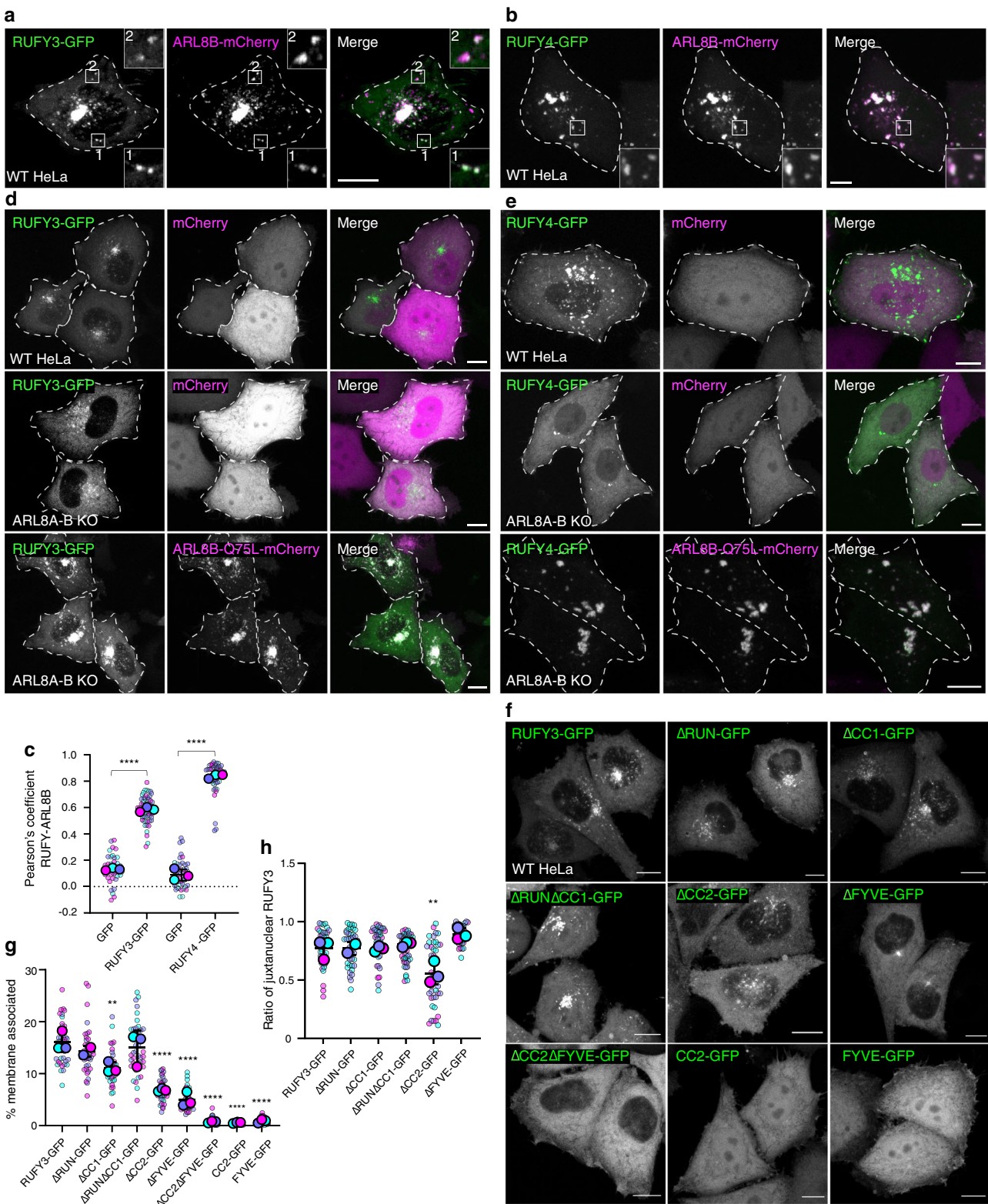

(Fig. 6e, g), and shifted the balance of the remaining LAMP1-RFP tracks to the retrograde direction (Fig. 6e, f) (for evidence of RUFY-LAMP1 co-movement, see also Supplementary Movie 1). In contrast, the velocity and run length of LAMP1-RFP vesicles were largely unaffected by expression of RUFY3-GFP or RUFY4-GFP (Fig. 6h, i).

Additional experiments using fixed neurons showed that RUFY3-GFP or RUFY4-GFP expression reduced the total number of endolysosomal vesicles labeled for endogenous LAMTOR4 in the axon (Supplementary Fig. 4a, b). Furthermore, live-cell and kymograph analyses showed that RUFY3-GFP or RUFY4-GFP perfectly co-moved with ARL8B-mCherry on axonal vesicles (Supplementary Fig. 4c, d), and that these vesicles moved mainly in the retrograde direction (Supplementary Fig. 4e, f). We also observed that most RUFY3-GFP-decorated vesicles were acidified, as shown by the high degree of co-staining with LysoTracker

**Fig. 3 Role of ARL8 and RUFY3 domains in association of RUFY3/4 with vesicles and juxtanuclear clustering of the vesicles. a, b** Live-cell imaging of HeLa cells co-expressing RUFY3-GFP or RUFY4-GFP (green) with ARL8B-mCherry (magenta). Dashed lines indicate cell edges. Scale bars: 10 μm. The insets are ~3-fold enlargements of the boxed areas. Single channels are shown in grayscale. Images are representative from three independent experiments with similar results. **c** SuperPlot representation[106] of the Pearson correlation coefficient for the co-localization of GFP (negative control), RUFY3-GFP or RUFY4-GFP with ARL8B-mCherry from experiments such as those in panel (**a**) and (**b**). Big circles represent the mean, and small dots the individual data points from each experiment. Experiments are color coded. Horizonal lines indicate the mean ± SD of the means from three independent experiments. Statistical significance was calculated using two-tailed unpaired Student's t test. **** p < 0.0001. **d, e** Live-cell imaging of WT or ARL8A-B–KO HeLa cells co-expressing RUFY3-GFP or RUFY4-GFP (green) and mCherry or ARL8B-Q75L-mCherry (magenta). Single channels are shown in grayscale. Scale bars: 10 μm. **f** Live-cell imaging of HeLa cells expressing RUFY3 deletion mutants (Fig. 2c) tagged with GFP. Images are in grayscale. Scale bars: 10 μm. **g** Cells with low-to-moderate expression of the RUFY3-GFP constructs in experiments such as that in panel (**f**) were selected for analysis of membrane recruitment. Total fluorescence intensity per cell was measured, and membrane-associated fluorescence of each construct was estimated by removing diffuse cytoplasmic signal with manual thresholding. Membrane-associated signal, as a percent of total cellular fluorescence, was recorded for 11–15 cells from each construct. Data were represented as SuperPlots as described for panel (**c**). Horizontal lines indicate the mean ± SD of the means from three independent experiments. Statistical significance was calculated using one-way ANOVA with multiple comparisons to RUFY3-GFP using Dunnett's test. **p < 0.01, ****p < 0.0001. **h** SuperPlot representation of the ratio of juxtanuclear GFP to total GFP calculated by shell analysis from experiments such as those in panel (**f**), represented as described for panel **c**. Horizontal lines indicate the mean ± SD of the means from three independent experiments. Statistical significance was calculated by one-way ANOVA with multiple comparisons to RUFY3-GFP using Dunnett's test. **p < 0.01.

(Supplementary Fig. 4g, h). The majority of these vesicles labeled for RUFY3-GFP and LysoTracker moved in the retrograde direction (Supplementary Fig. 4g, i, j), consistent with previous work showing that LysoTracker mainly stains retrograde endolysosomal vesicles in the axon[53]. In contrast to RUFY3-GFP vesicles, only about 40% of RUFY4-GFP vesicles were positive for LysoTracker (Supplementary Fig. 4g, h), although the majority also moved in retrograde direction (Supplementary Fig. 4g, i, j). This suggests that, in addition to acidic endolysosomes, RUFY4-GFP localizes to a distinct population of non-acidic vesicles.

Altogether, these results indicate that RUFY3 and RUFY4 promote retrograde transport of ARL8-positive endolysosomal vesicles from the axon to the soma, leading to a reduction in the number of axonal endolysosomes.

**Interaction of RUFY3 and RUFY4 with dynein-dynactin**. The phenotypes resulting from manipulation of RUFY3 and RUFY4 expression suggested that these proteins might play a role in transport driven by cytoplasmic dynein-dynactin, a microtubule motor involved in retrograde transport in the cytoplasm[54]. Indeed, we observed that both RUFY3-GFP and RUFY4-GFP co-immunoprecipitated with the endogenous dynein intermediate chain (DIC) and the endogenous p150^Glued subunit of dynactin in HEK293T cells, albeit to a lesser extent than the well-characterized dynein-dynactin adaptor protein BICD2[55] (Fig. 7a). In addition, purified, recombinant 6His-GFP-RUFY3, but not 6His-GFP (negative control), pulled down both endogenous DIC and p150^Glued from an extract of HEK293T cells, though also less well than 6His-GFP-BICD2_25-400 (the part of BICD2 that interacts with dynein-dynactin) (Fig. 7b). Recombinant 6His-GFP-RUFY4 was degraded during expression and purification, and could not be analyzed using this assay. Finally, we found that purified, recombinant 6His-GFP-RUFY3 could be pulled down with purified, recombinant GST-tagged dynein light intermediate chain 1 (DLIC1) and, more precisely, the C-terminal domain of DLIC1 (Fig. 7c), a domain that was previously implicated in interactions with other dynein adaptors[56,57]. These results thus indicated that RUFY3 and RUFY4 interact with dynein-dynactin, and that the interaction of RUFY3 involves the C-terminal domain of DLIC.

To test the functional relevance of interactions of RUFY3 and RUFY4 with dynein-dynactin, we compared the distribution of RUFY3-mCherry and RUFY4-mCherry in cells overexpressing the GFP-tagged CC1 domain of p150^Glued, a construct that functions as a dominant-negative inhibitor of dynein-dynactin[58] (Fig. 7d). We observed that overexpression of this construct prevented the redistribution of RUFY3-mCherry- and RUFY4-

mCherry-containing vesicles to the juxtanuclear area of the cell, instead causing localization of these vesicles to the cell periphery, with particular concentration at cell vertices (Fig. 7d, arrows). These observations demonstrated that interference with dynein-dynactin does not prevent the association of RUFY3 and RUFY4 with endolysosomes, but precludes their ability to move endolysosomes toward the cell center.

**Artificial targeting of RUFY3 and RUFY4 to peroxisomes promotes their juxtanuclear clustering in a dynein-dependent manner**. To determine whether RUFY3 and RUFY4 are sufficient to promote organelle coupling to dynein-dynactin, we next used a peroxisome re-localization assay[59]. Peroxisomes are particularly suited for this assay because they have a dispersed distribution and are not very motile. The assay consisted of co-expressing: (i) a peroxisomal targeting signal from PEX3 (amino acids 1–42) fused to FKBP and RFP, together with ii) RUFY3 or RUFY4 fused to FRB and GFP (Fig. 8a). As a positive control, we used a BICD2_25-400-FRB-GFP construct (Fig. 8a). The addition of rapalog brings together the FRB and FKBP domains, leading to the targeting of RUFY3 or RUFY4 to peroxisomes (Fig. 8b). We observed that, in the absence of rapalog, peroxisomes labeled with the PEX3_1-42-FKBP-RFP construct were scattered throughout the cytoplasm despite the co-expression with RUFY3-FRB-GFP, RUFY4-FRB-GFP or BICD2_25-400-FRB-GFP (Fig. 8c, e, -Rapalog). The addition of rapalog, however, resulted in the redistribution of PEX3_1-42-FKBP-RFP-labeled peroxisomes, together with RUFY3-FRB-GFP, RUFY4-FRB-GFP, or BICD2_25-400-FRB-GFP, to the juxtanuclear area of the cell (Fig. 8c, e, + Rapalog). In all cases, this redistribution was prevented by KD of the dynein heavy chain (DHC) (Fig. 8d, e, + Rapalog). These results thus demonstrated that artificial targeting of RUFY3 or RUFY4 to an unrelated organelle is sufficient to promote the dynein-dependent redistribution of this organelle toward the cell center.

**RUFY3 is required for juxtanuclear redistribution of endolysosomes upon cytosol alkalinization**. Cytosol alkalinization is well-known to cause redistribution of endolysosomes toward the cell center[60]. Indeed, we observed that, in HeLa cells treated with a non-targeting siRNA, alkalinization caused juxtanuclear clustering of LAMP1-containing endolysosomes (Fig. 9a, c). Treatment of HeLa cells with RUFY3 siRNA caused the expected dispersal of LAMP1-containing endolysosomes, including localization to cell vertices (Fig. 9b, c, arrows). In these cells, alkalinization failed to cause juxtanuclear clustering of endolysosomes,

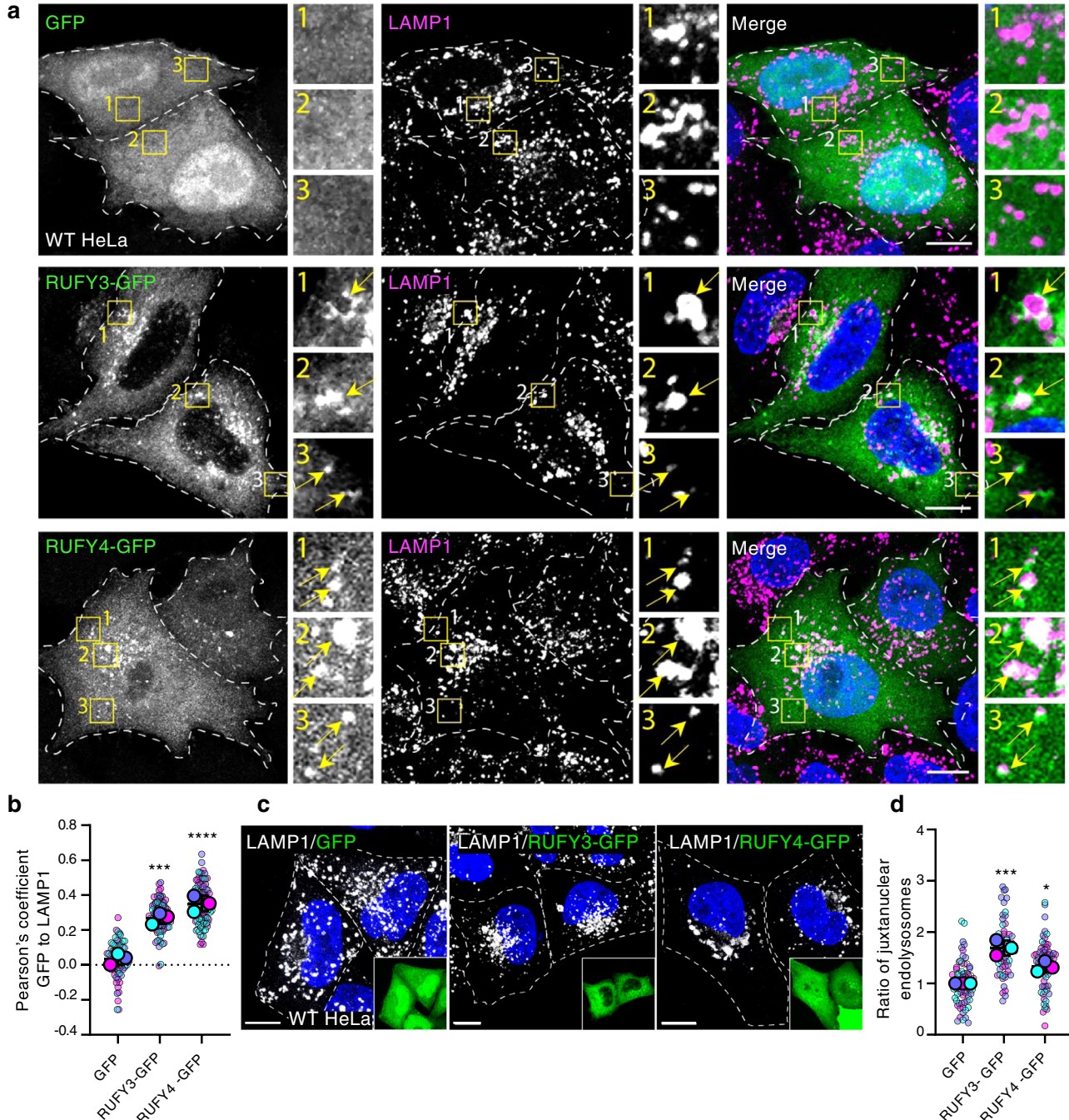

**Fig. 4 RUFY3 and RUFY4 localize to endolysosomes, and promote their juxtanuclear clustering. a** Co-localization of RUFY3-GFP and RUFY4-GFP with endogenous LAMP1. Immunofluorescence microscopy of HeLa cells transfected with plasmids expressing GFP (control), RUFY3-GFP or RUFY4-GFP (green), fixed and immunostained for endogenous LAMP1 (magenta). Nuclei were stained with DAPI (blue). Cells with low-to-moderate expression of RUFY3-GFP or RUFY4-GFP were selected for analysis of co-localization. Single channels are shown in grayscale. Cell edges are shown with dashed lines. Scale bars: 10 μm. Insets show 3-fold enlargements of the boxed areas. Arrows indicate vesicles where RUFY3/4-GFP proteins co-localize with LAMP1. **b** SuperPlot representation of the Pearson's correlation coefficient for the co-localization of GFP, RUFY3-GFP or RUFY4-GFP with endogenous LAMP1 from experiments such as that shown in panel (**a**). Values were calculated and represented as described for Fig. 3c. Horizontal lines indicate the mean ± SD of the means from three independent experiments. Statistical significance was calculated using one-way ANOVA with multiple comparisons to the GFP control using Dunnett's test. \*\*\**p* < 0.001, \*\*\*\**p* < 0.0001. **c** Overexpression of RUFY3-GFP or RUFY4-GFP causes juxtanuclear clustering of endolysosomes. This experiment was done as described for panel (**a**), except that highly overexpressing cells were chosen for analysis. Endogenous LAMP1 staining is shown in grayscale and GFP images in green (inset). The strong cytosolic staining of the GFP constructs is due to the overexpression. Nuclei were stained with DAPI (blue). Cell edges are highlighted with dashed lines. Scale bars: 10 μm. Insets show 2.85-fold reductions of the transfected cells. **d** SuperPlot representation of the ratio of juxtanuclear LAMP1 to total LAMP1 calculated by shell analysis from experiments such as those in panel (**c**). Values were calculated and represented as described for Fig. 3c, h. Horizontal lines indicate the mean ± SD of the means from three independent experiments. Statistical significance was calculated using one-way ANOVA with multiple comparison to the GFP control using Dunnett's test. \**p* < 0.05, \*\*\**p* < 0.001.

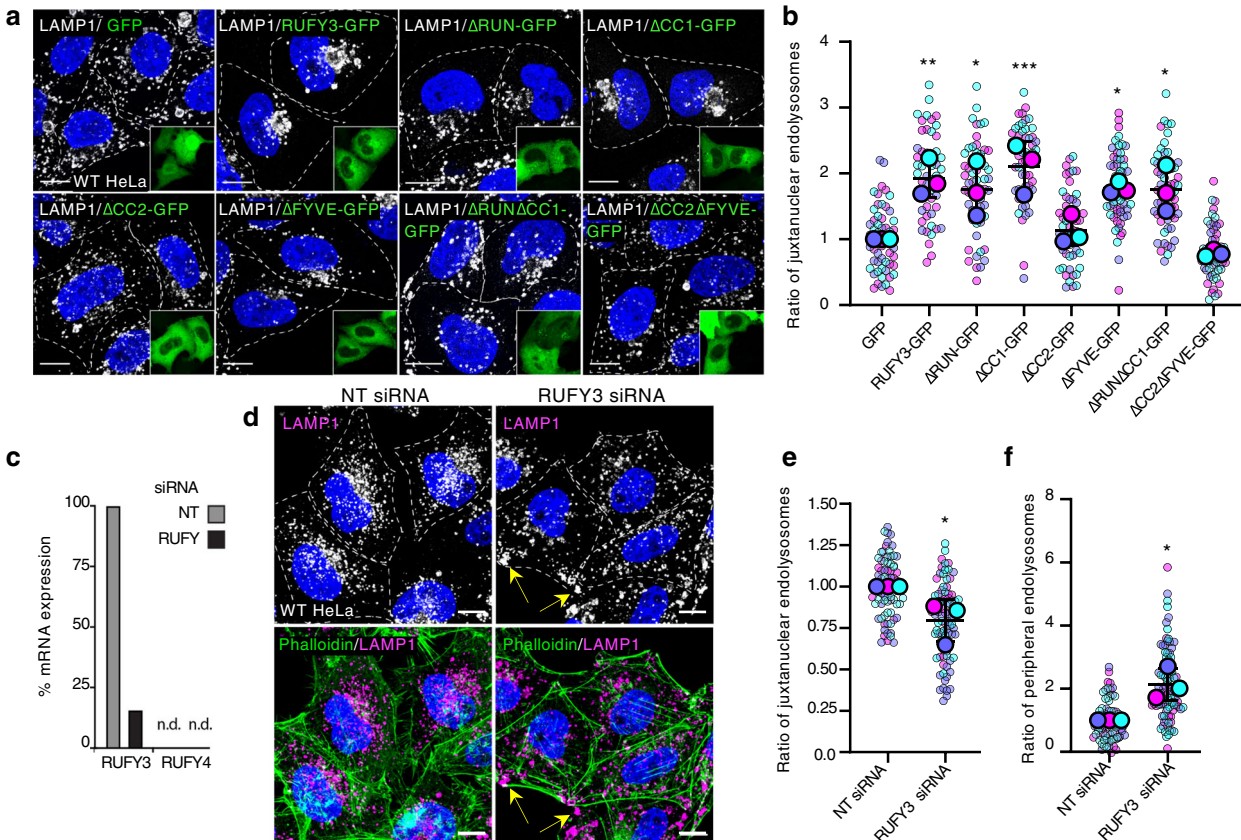

**Fig. 5 Effects of overexpressing RUFY3 deletion constructs or knocking down RUFY3 on the distribution of endolysosomes. a** Immunofluorescence microscopy of HeLa cells transfected with plasmids encoding GFP (control) or RUFY3-GFP deletion constructs depicted in Fig. 2c (green in the insets), fixed and immunostained for endogenous LAMP1 (grayscale). Nuclei were stained with DAPI (blue). Highly overexpressing cells were chosen for analysis, thus the strong cytosolic fluorescence of the constructs in the insets. Cell edges are highlighted with dashed lines. Scale bars: 10 μm. Insets show 2.85-fold reductions of the transfected cells. **b** SuperPlot representation as described for Fig. 3c, h of the effect of RUFY3-GFP deletion constructs on the distribution of LAMP1 from experiments such as that shown in panel (**a**). Horizontal lines represent the mean ± SD of the means from three independent experiments. Statistical significance was calculated using one-way ANOVA with multiple comparisons to GFP using Dunnett's test. * $p < 0.05$, ** $p < 0.01$, *** $p < 0.001$. **c** qRT-PCR of RUFY3 and RUFY4 mRNA relative to actin mRNA in HeLa cells treated with non-targeting (NT) or RUFY3/4 SMARTpool siRNAs. n.d., not detected. **d** Immunofluorescence microscopy of HeLa cells treated with non-targeting (NT) or RUFY3 SMARTpool siRNA and stained with antibodies to endogenous LAMP1 (grayscale and magenta) and Alexa fluor 546-conjugated phalloidin (green) to highlight cell edges. Nuclei were stained with DAPI (blue). Cell edges in grayscale images are highlighted with dashed lines. Yellow arrows indicate accumulation of endolysosomes at cell vertices. Scale bars: 10 μm. **e, f** SuperPlot representation as described for Fig. 3c, h of the effect of RUFY3 KD on the juxtanuclear (**e**) and peripheral (**f**) distribution of LAMP1 from experiments such as that shown in panel (**d**). Horizontal lines indicate the mean ± SD of the means from three independent experiments. Statistical significance was calculated using the two-tailed unpaired Student's *t* test. * $p < 0.05$. See also Supplementary Fig. 3.

although it did cause a slight displacement of endolysosomes toward the cell center (Fig. 9b, c). These experiments thus demonstrated that the function of RUFY3 in promoting transport of endolysosomes toward the cell center is at least partially required for changes in endolysosome positioning in response to changes in cytosolic pH.

## Discussion

At steady state, endolysosomes exhibit a characteristic cytoplasmic distribution, consisting of a densely packed population in the juxtanuclear area and a more scattered population in the peripheral area of the cell[27,61]. In polarized cells such as neurons, the peripheral population of endolysosomes includes distinct pools in specialized domains of the cells (e.g., axon and dendrites)[20,62–65]. The overall distribution of endolysosomes results from the integration of various processes, including tethering to other organelles such as the endoplasmic reticulum (ER)[66–69] and coupling to microtubule motors[70,71]. Transport of endolysosomes toward microtubule plus ends (i.e., anterograde transport) or minus ends

(i.e., retrograde transport) depends on coupling to kinesin or dynein-dynactin motors, respectively[70,71] (Fig. 9d). Coupling to both types of motor is often not direct but mediated by small GTPases, adaptors and other effectors and regulators[27,61]. Since there is only one cytoplasmic dynein (in contrast to the ~45 kinesins encoded in mammalian genomes), multiple combinations of adaptors and regulators allow coupling of dynein-dynactin not only to distinct organelles, but also to the same organelle with different functional properties. The multiple systems shown to couple endolysosomes to dynein-dynactin include the small GTPase RAB7 and its effector RILP[72], the transmembrane protein TMEM55B and adaptor protein JIP4[53,73], the related adaptor protein JIP3[53,74], the calcium channel MCOLN1 and penta-EF-hand protein ALG2[75], the septin protein SEPT9[76], the protein SNAPIN[77] and the sorting nexins SNX5 and SNX6 (for endolysosomal tubules)[78,79]. In the present study, we identify RUFY3 and RUFY4 as ARL8 effectors that promote coupling of endolysosomes to dynein-dynactin (Fig. 9d).

Previous studies had characterized a short, 469-amino-acid form of RUFY3 (denoted here as RUFY3.2), which lacks part of

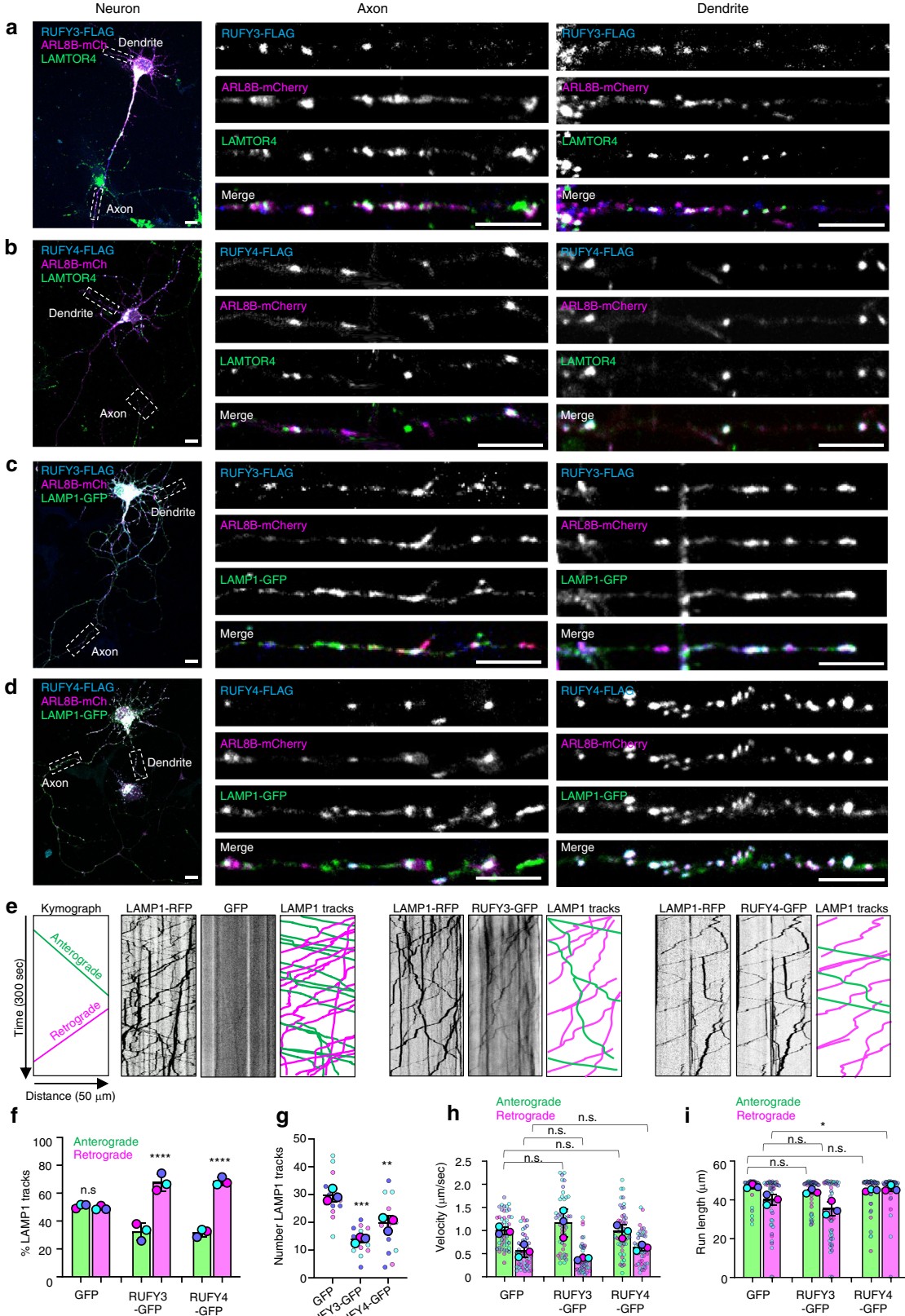

the CC2 domain and the entire FYVE domain present in the predicted long, 620-amino-acid form of the protein (RUFY3.1) (Fig. 1d). The short form had been shown to be particularly abundant in the brain, and to play roles in neuronal polarity and in the regulation of axon specification, growth and degeneration[40–43]. The existence, distribution, and function of the

long form had not been previously documented. Our MitoID procedure using ARL8A and ARL8B as baits identified RUFY3, including peptides only found in the longer RUFY3.1 form, as a top hit. This finding demonstrated that the longer form exists, and that it is expressed in non-neuronal cells. Together with expression data from the Human Protein Atlas (https://

**Fig. 6 RUFY3 and RUFY4 shift axonal endolysosome movement to the retrograde direction. a, b** Immunofluorescence microscopy of neurons transfected with plasmids encoding RUFY3-FLAG (**a**) or RUFY4-FLAG (**b**) and ARL8B-mCherry. Neurons were fixed, permeabilized, and RUFY-FLAG proteins detected by immunostaining with antibody to the FLAG epitope (blue), endolysosomes with antibody to endogenous LAMTOR4 (green), and ARL8B-mCherry by its intrinsic fluorescence (magenta). Images on the left show neurons (scale bars: 10 μm) with boxes indicating axons and dendrites that are enlarged on the right (scale bars: 5 μm). Images are representative from three independent experiments with similar results. **c, d** Same as panels (**a**) and (**b**), but neurons were co-transfected with a plasmid encoding LAMP1-GFP (green) instead of immunostained for LAMTOR4. Images are representative from three independent experiments with similar results. **e** Neurons were transfected with plasmids encoding LAMP1-RFP (magenta) along with GFP, RUFY3-GFP, or RUFY4-GFP (green), axons were imaged live, and trajectories of fluorescent particles were represented as kymographs. Single channels are represented in grayscale. Lines with negative or positive slopes in the kymographs correspond to vesicles moving in anterograde or retrograde directions, respectively. **f** Quantification of the percentage of anterograde (green) and retrograde (magenta) movement of LAMP1-RFP vesicles from experiments such as that in panel (**e**). Values are the mean ± SD from three independent experiments, with a total of 15 neurons and 445 (GFP), 206 (RUFY3-GFP), and 282 (RUFY4-GFP) LAMP1-RFP motile events analyzed per condition. Statistical significance was calculated using one-way ANOVA with multiple comparisons using Tukey's test. ****$p < 0.0001$; n.s., not significant. **g** SuperPlot representation of the total number of LAMP1-RFP tracks from experiments such as that in panel (**e**). Horizontal lines indicate the mean ± SD of the means from three independent experiments. Statistical significance was calculated using one-way ANOVA with multiple comparisons to the GFP control using Dunnett's test. **$p < 0.01$, ***$p < 0.001$. **h, i** SuperPlot representation of velocity and run length of LAMP1-RFP tracks in neurons from experiments such as in panel (**e**). The mean ± SD of the means from three independent experiments are indicated. Statistical significance was calculated using one-way ANOVA with multiple comparisons using Tukey's test. *$p < 0.05$; n.s., not significant. See also Supplementary Movie 1.

www.proteinatlas.org/search/rufy3), the isolation of RUFY3.1 from HEK293T cells is consistent with the additional involvement of RUFY3 in non-neuronal processes such as migration, invasion, and metastasis of lung, gastric and colorectal cancer cells[44–47].

The 571-amino-acid RUFY4 protein had been previously shown to be expressed mainly in lung and lymphatic organs, as well as in dendritic cells and macrophages[47]. The Human Protein Atlas also reports detectable expression of the RUFY4 mRNA in the brain, gastrointestinal tract and prostate (https://www.proteinatlas.org/search/rufy4), but very low levels in other tissues and cells, including the HeLa and HEK293T cell lines, and hippocampal neurons, used in our study. Functional studies revealed roles of RUFY4 in autophagosome formation, autophagosome-lysosome fusion and degradation of autophagic substrates such as damaged mitochondria and intracellular bacteria in phagocytic cells[50,80].

Our findings suggest that the functions of RUFY3 in neurons and cancer cells, and RUFY4 in phagocytic cells, might be related to the ability of these proteins to couple endolysosomes to dynein-dynactin. Indeed, processes such as the regulation of axonal functions[20,33,81], cancer cell migration, invasion and metastasis[31,32,82], and autophagy[10,15,20,29] have all been shown to be influenced by endolysosome positioning and motility, consistent with a role for RUFY3 and RUFY4 in the regulation of endolysosomal functions.

Further biochemical analyses confirmed that both RUFY3 and RUFY4 interact with the GTP-bound form of ARL8. Although ARL8 was previously shown to bind to the RUN domains of SKIP and PLEKHM1[10,11,18,20], we found that binding of ARL8 to RUFY3 involves the CC2 domain of RUFY3. These observations imply that ARL8 can bind its effectors by different mechanisms. Both ARL8 and the CC2 domain were found to contribute partially to the association of RUFY3 with vesicles, and critically to the clustering of the vesicles in the juxtanuclear area of the cell. The FYVE domain also contributed partially to the association of RUFY3 with vesicles, though not to their juxtanuclear clustering. It remains to be established how the RUFY3-FYVE domain mediates this association, since it lacks the tandem histidine cluster required for binding to PtdIns(3)P on endolysosomal membranes[37]. Because RUFY4 is not endogenously expressed in the cells used in this study, we did not perform detailed structure-function analyses of this protein. However, analysis of transgenic RUFY4 localization revealed that ARL8 is even more important for the association of this protein with endolysosomes.

Despite having homology to RUFY3 in the region of the CC2 domain and other domains, RUFY1 and RUFY2 did not interact

with ARL8. Instead, RUFY1 was previously shown to interact with the small GTPases RAB4, RAB5, and RAB14, and to regulate early endosomal functions[83–86]. RUFY2, on the other hand, was shown to interact with the Golgi complex-associated small GTPase RAB33A, which functions in autophagosome formation[87,88]. These interactions and functions are consistent with the differences in the association of ARL8 with different RUFY family members.

In both HeLa cells and rat hippocampal neurons, transgenic RUFY3 or RUFY4 constructs co-localized with ARL8 and LAMP1 on endolysosomes. Moreover, overexpression of RUFY3 or RUFY4 constructs in HeLa cells caused juxtanuclear clustering of endolysosomes. This effect was dependent on ARL8 and the RUFY3-CC2 domain, thus providing a functional correlate for the ARL8-CC2 domain interaction. Moreover, RUFY3 or RUFY4 overexpression in neurons caused a shift toward retrograde transport, and thereby a reduction in the number, of axonal endolysosomes. Although these effects of RUFY3/4 over-expression could be due to competition of the interaction of ARL8 with SKIP–kinesin-1 and kinesin-3 motor systems, several lines of evidence suggest that the effects are more directly related to the ability of RUFY3/4 to couple endolysosomes to dynein-dynactin. First, KD of RUFY3 in HeLa cells resulted in dispersal of endolysosomes toward the cell periphery. Furthermore, RUFY3 and RUFY4 interact with dynein-dynactin. Lastly, artificial attachment of RUFY3 and RUFY4 to peroxisomes promote their dynein-dynactin–dependent redistribution toward the cell center. These effects are most consistent with a role for RUFY3 and RUFY4 in coupling of endolysosomes to dynein-dynactin, and for the maintenance of the juxtanuclear population of endolysosomes at steady state.

Co-immunoprecipitation and pull-down analyses showed that RUFY3 and RUFY4 physically interact with dynein-dynactin. Pull-down experiments with all-recombinant proteins, in particular, suggested a direct interaction of RUFY3 with the DLIC subunit of dynein, a property shared with known dynein-dynactin activating adaptors such as BICD2, SPDL1, and HOOK1-3[54]. However, the interaction of RUFY3 with dynein-dynactin was weaker than that of BICD2, raising the possibility that RUFY3 exerts its functions in cooperation with other adaptors.

The function of RUFY3 and RUFY4 in retrograde transport of endolysosomes is surprising in light of the many other proteins that were previously shown to couple endolysosomes to dynein-dynactin. A possible explanation for the existence of multiple adaptors is that they all contribute to the overall strength of

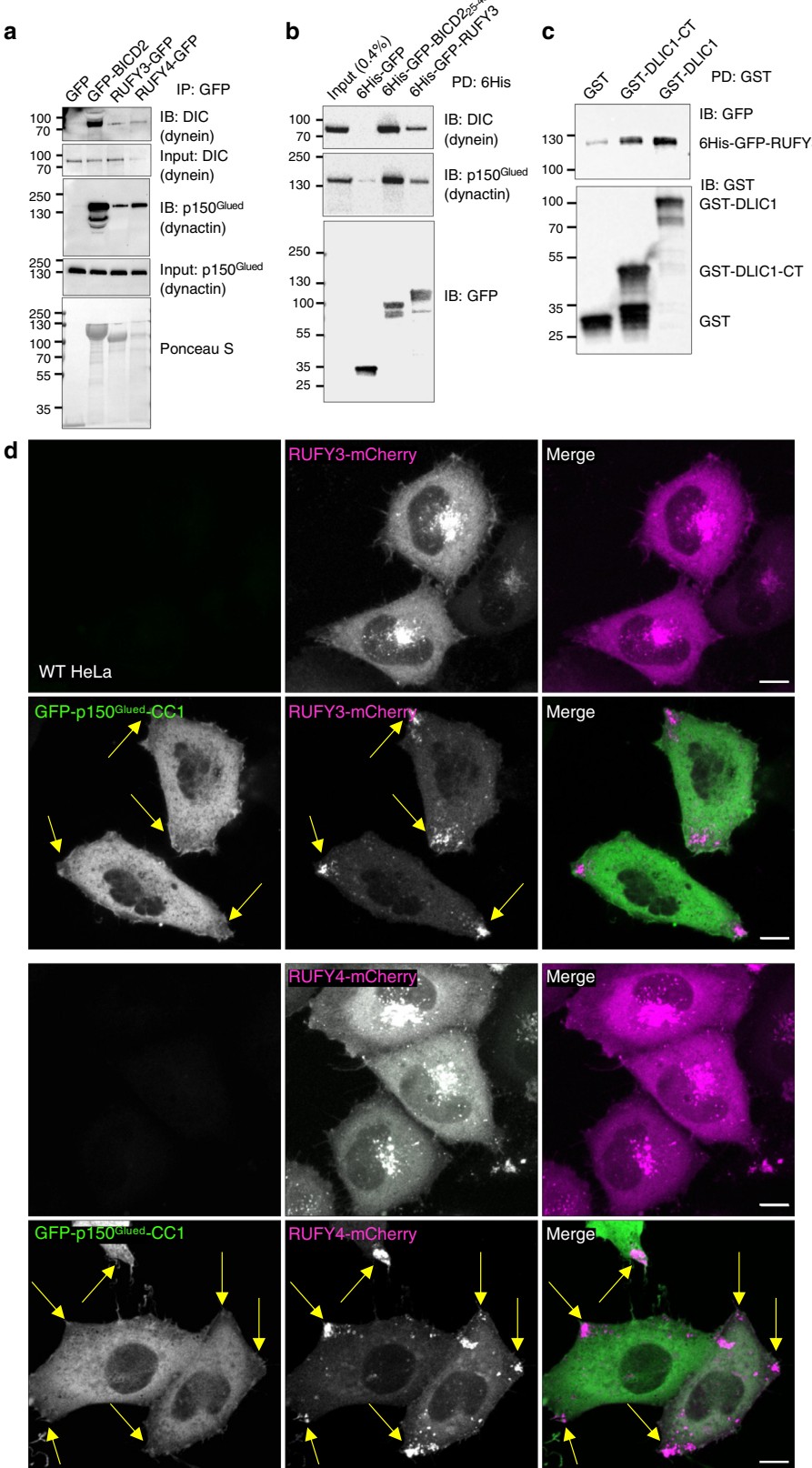

coupling. The absence of any of these adaptors could weaken the interactions with dynein-dynactin, tilting the balance toward interactions with kinesins and thus shifting the distribution of endolysosomes toward the cell periphery. The different dynein-dynactin adaptors could also have cell-type specific functions, depending on their relative expression levels in different cells

(e.g., RUFY4 in phagocytic cells). In addition, various dynein-dynactin adaptors could be differentially regulated in response to specific stimuli, as would be expected from their interactions with different GTPases and calcium-binding proteins. Furthermore, the adaptors could be associated with different populations or domains of endolysosomes. For example, RUFY3 and RILP

**Fig. 7 Interaction of RUFY3 and RUFY4 with dynein-dynactin. a** HEK293T cells were transfected with plasmids encoding GFP (negative control), GFP-BICD2 (positive control), RUFY3-GFP or RUFY4-GFP. Cell extracts were analyzed by immunoprecipitation (IP) with antibody to GFP, followed by immunoblotting (IB) for endogenous dynein intermediate chain (DIC) and endogenous p150[Glued] subunit of dynactin. Ponceau S staining shows the levels of immunoprecipitated GFP-tagged proteins. The experiment shown in this panel is one of two with similar results. **b** Extracts of HEK293T cells were incubated with recombinant 6His-StrepII-sfGFP (abbreviated 6His-GFP in the figure) (negative control), His6-StrepII-sfGFP-BICD2[25-400] (His6-GFP-BICD2[25-400]) (positive control) or 6His-StrepII-sfGFP-RUFY3 (6His-GFP-RUFY3), pulled down (PD) with Strep-Tactin agarose, and immunoblotted for endogenous dynein intermediate chain (DIC), the endogenous p150[Glued] of dynactin, or GFP. The GFP used to make these constructs is a variant named sfGFP, for super-folder GFP. The experiment shown in this panel is one of two with similar results. **c** Glutathione-Sepharose preloaded with purified, recombinant GST (negative control), GST-DLIC1 or GST-DLIC1-CT (C-terminal domain) was incubated with purified, recombinant 6His-StrepII-sfGFP-RUFY3 (6His-GFP-RUFY3). Bound proteins were detected by immunoblotting with antibodies to GFP and GST. The positions of molecular mass markers (in kDa) in panels **a-c** are indicated at left. **d** Live-cell images of HeLa cells co-expressing RUFY3-mCherry or RUFY4-mCherry (magenta) without or with GFP-p150[Glued]-CC1 (green). Single-channel images are shown in grayscale. Scale bars: 10 μm. Arrows point to RUFY proteins at cell tips. This experiment is one of two with similar results.

likely associate with ARL8- and RAB7-decorated endolysosomes, respectively[72] (this study), and SNX5 and SNX6 associate with endolysosomal tubules[78]. Finally, different dynein-dynactin adaptors could participate in a sequential handoff mechanism, as recently reported for the retrograde transport of maturing autophagosomes in the axon[89].

Another conundrum that remains to be solved is how ARL8 can regulate both anterograde endolysosome transport through recruitment of kinesin-1 and kinesin-3[11,17], and retrograde endolysosome transport through recruitment of dynein-dynactin (this study) (Fig. 9d). Moreover, studies in Drosophila showed that ARL8 can also interact with the ortholog of RILP[21], a known dynein-dynactin interacting protein[72]. This regulation of opposing processes by the same GTPase is not exclusive to ARL8, however, given that RAB7 also promotes anterograde endolysosome transport via FYCO1[67] and retrograde endolysosome transport via RILP[72]. For both GTPases, there must be other regulators that determine the interaction with alternative adaptors and, consequently, the direction of endolysosome transport. In any event, the role of ARL8 in anterograde transport seems to be dominant over that in retrograde transport, since depletion of ARL8 or its positive regulator BORC cause juxtanuclear clustering of endolysosomes, whereas overexpression of ARL8 drives endolysosomes to the cell periphery[7,11,17,18,20,29,64]. Future studies will have to address under what conditions ARL8 promotes endolysosome retrograde transport dependent on RUFY3 and RUFY4.

Additional modifiers of the RUFY3 function could include the small GTPases RAP2 and RAB33 via interactions with the RUN and CC1 domains, respectively[87,90]. The shorter RUFY3.2 species also interacts with RAB33 via the RUN domain[87]. RUFY4 was shown to interact with RAB7 through its RUN domain[50], although in our experiments we did not observe redistribution of RUFY4-GFP to mitochondria in cells expressing mitochondrially targeted RAB7A-Q67L-BirA*-HA-MAO. The significance of these interactions for the role of RUFY3 and RUFY4 in endolysosomal positioning and transport remains to be addressed.

While this manuscript was under preparation, a preprint was posted also reporting that RUFY3 mediates the interaction of ARL8B-GTP with dynein-dynactin for the localization of lysosomes to the juxtanuclear area[91]. Biochemical experiments by these authors demonstrated that RUFY3 physically interacts with JIP4 as a requisite for its function in lysosome positioning, providing direct evidence for the cooperation of these two types of dynein-dynactin adaptors/regulators[91].

Further studies will be needed to elucidate how the function of multiple endolysosomal dynein-dynactin adaptors is integrated and how these functions are coordinated with those of kinesin adaptors to control the dynamic distribution of endolysosomes under different physiological and pathological conditions.

## Methods

**Recombinant DNAs.** ARL8 is normally anchored to the endolysosome membrane via an N-terminal α-helix[4]. To mimic this topology, mitochondrially targeted ARL8 (Mito-ARL8) constructs used for MitoID were created by fusing a mitochondrial-targeting sequence (MTS) and the BioID2 biotin ligase to the N-terminus of ARL8. To this end, cDNA sequences encoding the mitochondrial-targeting sequence (MTS) of human TOM20 (amino acids 1–30, MVGRNSAIAAGVCGALFIGYCIYFDRKRRS)[38], followed by a short GAGA linker, were inserted into the pcDNA3.1-myc-BioID2-MCS plasmid[39] (a gift from Kyle Roux, Addgene #74223) by PCR to create pcDNA3.1-TOM20-MTS-myc-BioID2. Next, cDNA sequences encoding human ARL8A or ARL8B lacking the N-terminal α-helix (amino acids 1-17) and harboring the Q75L or T34N mutations, and an N-terminally GAGA linker, were inserted into the XhoI and BamHI sites of pcDNA3.1-TOM20-MTS-myc-BioID2. The resulting plasmids encoded TOM20-MTS-GAGA-myc-BioID2-GAGA-ARL8 fusion proteins (Mito-BioID2-ARL8). Plasmids encoding RUFY3 deletion mutants were generated by KLD mutagenesis (Cat# M0554S, New England Biolabs) on the backbone of RUFY3-GFP and RUFY3-FLAG plasmids (see below). The plasmid pcDNA3.1-SKIP-FOS was generated by insertion of SKIP coding sequences into the XbaI and KpnI sites of pcDNA3.1-FOS (FLAG-One-Strep). DNA sequences encoding the peroxisome-targeting sequence of PEX3 (amino acids 1-42) were cloned by KLD mutagenesis into the pEGFP-N1-SKIP[1-300]-FKBP-mRFP[18] vector to create pEGFP-N1-PEX3[1-42]-FKBP-mRFP.

RUFY species used in this study were RUFY1 isoform 1 (NM_025158.5, https://www.ncbi.nlm.nih.gov/nuccore/NM_025158.5/), RUFY2 isoform 1 (NM_017987.4, https://www.ncbi.nlm.nih.gov/nuccore/NM_017987.4), RUFY3 isoform 1 (RUFY3.1) (NM_001037442.4, https://www.ncbi.nlm.nih.gov/nuccore/NM_001037442.4), RUFY3 isoform 2 (RUFY3.2) (NM_014961.5, https://www.ncbi.nlm.nih.gov/nuccore/NM_014961.5), and RUFY4 isoform 1 (NM_198483.3, https://www.ncbi.nlm.nih.gov/nuccore/NM_198483.3). pcDNA3.1 + /C-(K)-DYK-RUFY-FLAG plasmids OHu19866D, OHu02933D, OHu24594D, OHu24610D, OHu55786D, respectively, were purchased from GenScript Biotech. These plasmids were used to create plasmids encoding RUFY-GFP and RUFY-mCherry constructs by amplifying RUFY coding sequences and inserting them into EcoRI-digested pEGFP-N1 and pmCherry-N1 plasmids, respectively, by Gibson assembly[92].

To create a pEGFP-N1-RUFY3-FRB-EGFP, a pEGFP-N1-SKIP[1-300]-FRB-EGFP[18] plasmid was digested with SalI and AgeI, and the fragment containing the FRB coding sequence was cloned into pEGFP-N1-RUFY3-GFP digested with the same enzymes. To create a pEGFP-N1-RUFY4-FRB-EGFP plasmid, pEGFP-N1-SKIP[1-300]-FRB-EGFP[18] was digested with XhoI and SalI and the fragment containing the FRB coding sequence was cloned into pEGFP-N1-RUFY4-EGFP digested with the same enzymes. A BICD2 fragment encoding amino acids 25-400 was amplified by PCR from mCh-BICD2*-Strep[93] (previously made in our lab and available from Addgene #120168), digested with BamHI and SalI, and ligated into pEGFP-N1-SKIP[1-300]-FRB-EGFP[18] digested with BglII and SalI. To create pET28a-6His-StrepII-sfGFP-RUFY3.1, the coding sequence of RUFY3.1 was amplified by PCR and inserted into KpnI and NotI double-digested pET28a-6His-StrepII-sfGFP-BICD2 (to replace BICD2 with RUFY3) by Gibson assembly. To create pEGFP-C1-p150[Glued]-CC1, the cDNA sequence encoding the CC1 domain (amino acids 205-540) from chicken p150[Glued] was cloned into the pEGFP-C1 plasmid between and EcoRI and SalI sites. To create pGEX6P-3-GST-LIC1[69], the LIC sequence was amplified by PCR, digested with BamHI and XhoI, and ligated into pGEX6P-3 that was digested with the same enzymes. All oligonucleotide primers are listed in Supplementary Table 2.

Other plasmids used in our study were: pMSCV-N-FLAG-HA-HOOK[94] (gift from Wade Harper), pLAMP1-RFP[95] (gift from Walter Mothes, Addgene #1817), and pEGFP-C1-FLAG[96] (gift from Steve Jackson, Addgene# 46956), GFP-BICD2[97] (gift from Anna Akhmanova), pET28a-6His-StrepII-sfGFP-BICD2[98] (gift from Ron Vale), pOPINE-GFPnanobody[99] (gift from Brett Collins, Addgene #49172), GFP-RILP[100] (gift from Cecilia Bucci), and RAB7A-Q67L-BirA*-HA-MAO and RAB7A-T22N-BirA*-HA-MAO[36] (gifts from Sean Munro, Addgene #128904 and #128905). All plasmid sequences were verified by Sanger sequencing (Genewiz or Eurofins Genomics). All plasmids are listed in Supplementary Table 3.

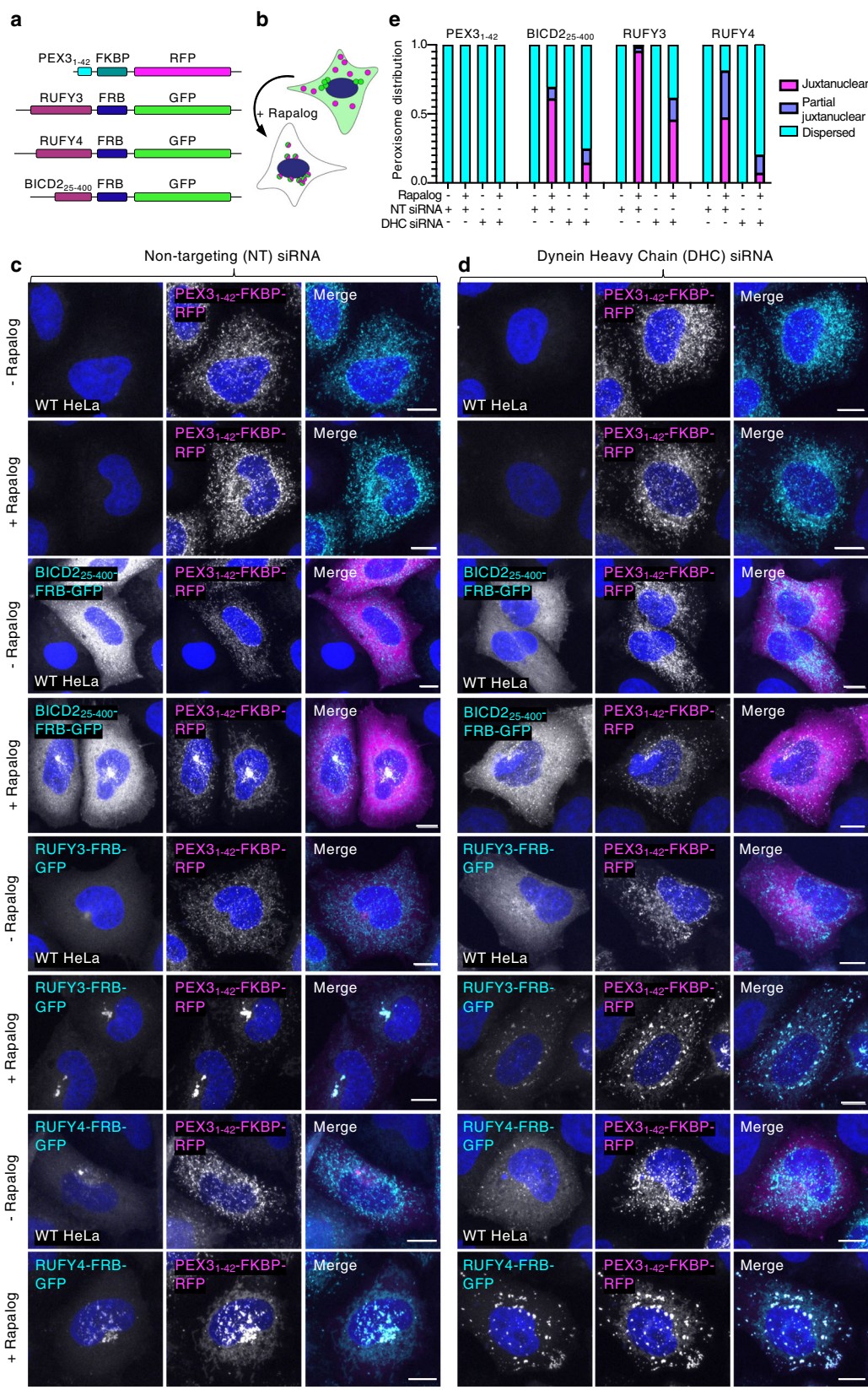

**Cell culture and treatments.** HeLa (Cat# CCL-2, ATCC) and HEK293T (Cat# 632180, Takara Bio) cells were maintained in Dulbecco's Modified Eagle's Medium (DMEM) (Cat# 112-319-101, Quality Biological) with 10% fetal bovine serum (35-011-CV, Corning), 50 U/mL penicillin, 50 µg/mL streptomycin (Cat# 30002-CL, Corning) (CDMEM) and incubated in 5% $CO_2$ and 37 °C. Lipofectamine 2000 (Cat# 11668019, Thermo Fisher) was used for transfections according to the manufacturer's protocol. Briefly, for immunofluorescence microscopy and live-cell

imaging, 0.1–0.5 µg plasmid with 1 µl Lipofectamine was used for transfection in 24-well and live-cell imaging chambers. 100 µl transfection mixture in Opti-MEM (Cat# 31985070, Gibco) was added to wells with 400 µl fresh CDMEM. Culture medium was replaced by CDMEM 1 h after transfection. Cells were fixed or imaged ~24 h after transfection. For co-immunoprecipitation experiments, 1–8 µg plasmid DNA and 25 µl Lipofectamine was used per 10 cm plate. A 3 mL transfection

**Fig. 8 Targeting of RUFY3 and RUFY4 to peroxisomes causes dynein-dependent re-localization of peroxisomes to the juxtanuclear area. a** Schematic representation of constructs used in the peroxisome re-localization assay. PEX3₁₋₄₂: peroxisomal-targeting signal from PEX3; FKBP: FK506-binding protein; FRB: FKBP rapamycin binding. Constructs are represented in N- to C-terminal direction. BICD2₂₅₋₄₀₀ was used as a positive control for a known dynein-dynactin adaptor domain. FKBP binds to FRB upon addition of rapalog. **b** Schematic representation of the rapalog-induced juxtanuclear re-localization of peroxisomes labeled by PEX3₁₋₄₂-FKBP-RFP (magenta) by a hypothetical dynein-dynactin adaptor fused to FRB and GFP (green). **c, d** Fluorescence microscopy of HeLa cells treated with non-targeting (NT) siRNA (c) or dynein heavy chain (DHC) siRNA (d), co-transfected with plasmids encoding the indicated proteins, and incubated for 1 h without (-) or with (+) 0.5 μM rapalog. Nuclei were stained with DAPI. Scale bars: 10 μm. This experiment is representative of 3 experiments with similar results. **e** Graph showing the fractional peroxisome distribution in the experiments shown in panels **c, d**. A minimum of 200 cells from two to three independent experiments were visually scored for the distribution of peroxisomes (fraction of cells with juxtanuclear, partially juxtanuclear, and dispersed peroxisomes).

mixture in Opti-MEM was added to plates containing 12 mL fresh CDMEM. Cells were harvested ~24 h after transfection.

The following siRNAs were used: non-targeting siRNA (5′-UGGUUUACAUG UCGACUAAUU-3′ (Dharmacon) (labeled with phosphate at the 5′), ON-TAR GETplus Human RUFY3 siRNA SMARTpool (Cat# L-020336-00-0005, Horizon Discovery), and the individual siRNAs (Cat# LQ-020336-00-0005, Horizon Discovery), Silencer Select siRNA to DYNC1H1 (ID: s4200, Cat# 4390824, Thermo Fisher). siRNA treatments were done with Oligofectamine (Cat# 12252011, Thermo Fisher) according to the manufacturer's protocol. Briefly, 2.5 μl of 20 μM siRNA was used per 24-well plate, or 10 μl per 6-well plate. For Fig. 5d, one shot of siRNA was used in a 48-h treatment. For Fig. 8, cells were treated with one shot of siRNA and transfected with plasmids 24 h after the siRNA shot. The peroxisome positioning assay was carried out 24 h after transfection (a total of 48 h siRNA treatment). The cells were treated with or without 0.5 μM rapalog (Cat# 635057, Takara Bio) for 1 h. For siRNA experiments in Fig. 9a, b, HeLa cells were treated with the siRNAs for 96 h (two shots of siRNA). Alkaline medium treatment was performed by incubating cells for 1 h at 37 °C in complete DMEM adjusted to pH 8.5 with NaOH. After incubation, cells were fixed with 4% paraformaldehyde for 15 min at room temperature and processed for immunofluorescence microscopy.

Cover slips and live-cell chambers were pre-coated with fibronectin (Cat# F2006, Millipore-Sigma). The following plates were used in the study: 4- and 8-well live-cell chambers (Cat# C4-1.5H-N, Cat# C8-1.5H-N, Cellvis), 10-cm plates (Cat# 353003, Corning), 15-cm plates (Cat# 353025, Corning) and 24-well plates (Cat# 353047, Corning).

**Proximity biotinylation using MitoID.** ARL8-interacting proteins were identified by MitoID[36,101] with modifications. HEK293T cells (5.4 × 10⁶) were plated on 15-cm plates (Cat# 353025, Corning). The next day, cells were transfected with 50 μl Lipofectamine 2000 (Cat# 11668019, Thermo Fisher) and 25 μg plasmid encoding Mito-ARL8 constructs and Mito-BioID2 (negative control) (Fig. 1a). We prepared two 15-mL tubes with Opti-MEM; one was mixed with the DNA and the second with Lipofectamine 2000. After 5-min incubation at room temperature, the contents of the tubes were combined, and the mix incubated at room temperature for an additional 20 min. The 6 mL mix was added to the cells on plates that were filled with 24 mL of fresh, prewarmed CDMEM supplemented with MycoZap Plus-CL (Cat# VZA-2011, Lonza). At 22 h after transfection, 50 μM biotin (Cat# 47868, Millipore-Sigma) was added to each plate (1.5 mL from 1 mM stock). At 24 h after biotin addition, cells were scraped from the plate in 4 mL cold PBS and washed 3 times with centrifugation for 5 min, at 4 °C, 500 × g. Cell pellets were kept at −80 °C. Two plates were used for each condition. The experiment was done with 3 biological replicates, and all samples were processed simultaneously. Thawed cells were resuspended in 5 mL buffer A (25 mM Tris-HCl pH 7.4, 150 mM NaCl, 1 mM EDTA 1% Triton X-100) supplemented with a protease inhibitor tablet (Cat# 1836170, Roche). The two plates corresponding to the same condition were combined at this stage and incubated for 1 h, at 4 °C with gentle rotation. The soluble fraction was separated by centrifugation for 20 min at 4 °C, 17,000 × g. A NeutrAvidin-agarose slurry (Cat# 29201, Pierce™ NeutrAvidin™ Agarose) (500 μl, corresponding to 250 μl beads) was washed in 14 mL buffer A. The supernatant was incubated with the NeutrAvidin-agarose overnight at 4 °C with gentle rotation. The beads were separated from the lysate by centrifugation for 5 min at 500 × g and 4 °C, and washed twice in 3 mL buffer B (2% SDS), 3 times in 5 mL buffer C (0.1% deoxycholic acid, 1% Triton X-100, 1 mM EDTA, 0.5 M NaCl, 50 mM HEPES pH 7.5), and once in 5 mL 50 mM Tris-HCl pH 7.4, 50 mM NaCl. Between washes, samples were centrifuged for 5 min at 4 °C, 500 × g. Lastly, the washed NeutrAvidin-agarose was resuspended in 75 μl 4X Laemmli buffer (Cat# 1610747, Bio-Rad) and samples were heated for 10 min at 99 °C. 60 μl were loaded onto 12% TGX precast gels (Cat# 4561043, Bio-Rad), which were run for a few minutes to allow the sample to enter the gel.

**Mass spectrometry.** Bands containing the entire sample were cut from the gel. Samples were reduced with 10 mM TCEP for 1 h, alkylated with 10 mM NEM for 10 min, and digested with trypsin at 37 °C overnight. Peptides were extracted from the gel and desalted using Oasis HLB μElution plates (Waters). Digests of each sample were injected into an Ultimate 3000 RSLC nano HPLC system (Thermo

Fisher). Peptides were separated on an ES802 column over a 66-min gradient with mobile phase B (98% acetonitrile, 1.9% H₂O, 0.1% formic acid) increased from 5% to 24%. LC-MS/MS data were acquired on an Orbitrap Lumos mass spectrometer (Thermo Fisher Scientific) in data-dependent acquisition mode. The MS1 scans were performed in Orbitrap with a resolution of 120 K, a mass range of 375–1500 m/z, and an AGC target of 2 × 10⁵. The quadrupole isolation was used with a window of 1.5 m/z. The MS/MS scans were triggered when the intensity of precursor ions with a charge state between 2 and 6 reached 1 × 10⁴. The MS2 scans were conducted in ion trap. The CID method was used with collision energy fixed at 30%. The instrument was run in top speed mode. MS1 scan was performed every 3 sec, and as many MS2 scans were acquired within the 3 s cycle. Database search and label-free quantification were performed using Proteome Discoverer 2.4 software. Up to 2 missed cleavages were allowed for trypsin digestion. NEM on cysteines and oxidation on methionine were set as fixed and variable modifications, respectively. Mass tolerances for MS1 and MS2 scans were set to 10 ppm and 0.6 Da, respectively. The search results were filtered by a false discovery rate of 1% at the protein level. Sequest HT was used for database search. Raw data were searched against the Sprot Human Canonical database. Percolator was used for peptide spectrum match validation. The summed intensity of the unique peptides was used for protein ratio calculation, with no imputation for missing values. The maximum and minimum fold changes allowed were set to 100 and 0.01, respectively. The default value of 100 was set as the maximum fold change allowed. For example, if the calculated ratios are 50, 80, 120, and 150 for proteins A, B, C, and D, the ratios reported by Proteome Discoverer software are 50, 80, 100, 100 for proteins A, B, C, and D. The abundance ratio was not transformed in any way. It was plotted on the graph as presented in the Excel file. The individual protein ANOVA method was used for hypothesis testing. Proteins with log2 fold change ≥ 1 or ≤ −1, and adjusted p ≤ 0.05, were considered significantly changed.

**Antibodies.** Primary antibodies (catalog numbers, names, animal species, working dilutions and sources in parentheses): FLAG-HRP (Cat# A8592, RRID:AB_439702, mouse, 1:5,000-1:6,000, Millipore-Sigma), ARL8A (Cat# 17060-1-AP, RRI-D:AB_2058998, rabbit, 1:500, Proteintech), ARL8B (Cat# 13049-1-AP, RRI-D:AB_2059000, rabbit, 1:500, Proteintech), TOM20 (Cat# 11802-1-AP, RRID:AB_2207530, rabbit, 1:500, Proteintech), BioID2 (Cat# BID2-CP-100, chicken, 1:2000, BioFront Technologies), p150^Glued (Cat# 610473, RRI-D:AB_397845, mouse, 1:300, BD Biosciences), DIC (Cat# MAB1618, RRID: AB_2246059, mouse,1:200, Millipore-Sigma), Streptavidin-HRP (Cat# 21130, 1:10,000, Pierce), GFP-HRP (Cat# 130-091-833, RRID:AB_247003 mouse, 1:2,000, Miltenyi Biotec), LAMTOR4 (C7orf59) (D4P6O) (Cat# 13140, RRID:AB_2798129, rabbit, 1:200, Cell Signaling Technology), LAMP1 (DSHB Hybridoma Product H4A3, mouse, 1:500, deposited by J.T. August and J.E.K. Hildreth), FLAG (Cat# F1804, RRID:AB_262044, mouse, 1:200, Millipore-Sigma), RUFY3 (Cat# NBP1-89614, RRID:AB_11022810, rabbit, 1:500, Novus Biological). HA (Cat# 11867423001, RRID: AB_390918, rat, 1:300, Roche). Pan-Neurofascin extracellular (Cat# A12/18, RRID:AB_2877334, mouse, 1:100, UC Davis/NIH NeuroMab Facility).

Secondary antibodies: HRP-conjugated goat anti-rabbit IgG (H + L), (Cat# 111-035-003, RRID:AB_2313567, 1:10,000, Jackson ImmunoResearch), HRP-conjugated donkey anti-mouse IgG (H + L) (Cat# 715-035-150, RRID:AB_2340770, 1:10,000, Jackson ImmunoResearch), donkey-anti-mouse IgG Alexa Fluor 488 (Cat# A21202, RRID:AB_141607, 1:2,000, Thermo Fisher), donkey-anti-mouse IgG Alexa Fluor 555 (Cat# A31570, RRID:AB_2536180, 1:2000, Thermo Fisher), goat anti-Chicken IgY (H + L) Alexa Fluor 555 (Cat# A21437, RRID:AB_1500593, 1:1000, Thermo Fisher), donkey anti-mouse IgG Alexa Fluor 647 (Cat# A31571, RRID:AB_162542, 1:1,000, Thermo Fisher). We also used Alexa Fluor 546-phalloidin (Cat# A22283, 1:2000, Thermo Fisher) and Alexa Fluor 633-phalloidin (Cat# A22284, 1:400, Thermo Fisher).

**Immunofluorescence microscopy.** Cells were washed 3 times with PBS, fixed with 4% paraformaldehyde (PFA) for 15 min at room temperature, washed 3 times with PBS, incubated with PBS supplemented with 0.1% saponin and 0.5–1% BSA (blocking buffer) for 30 min at room temperature, incubated with primary

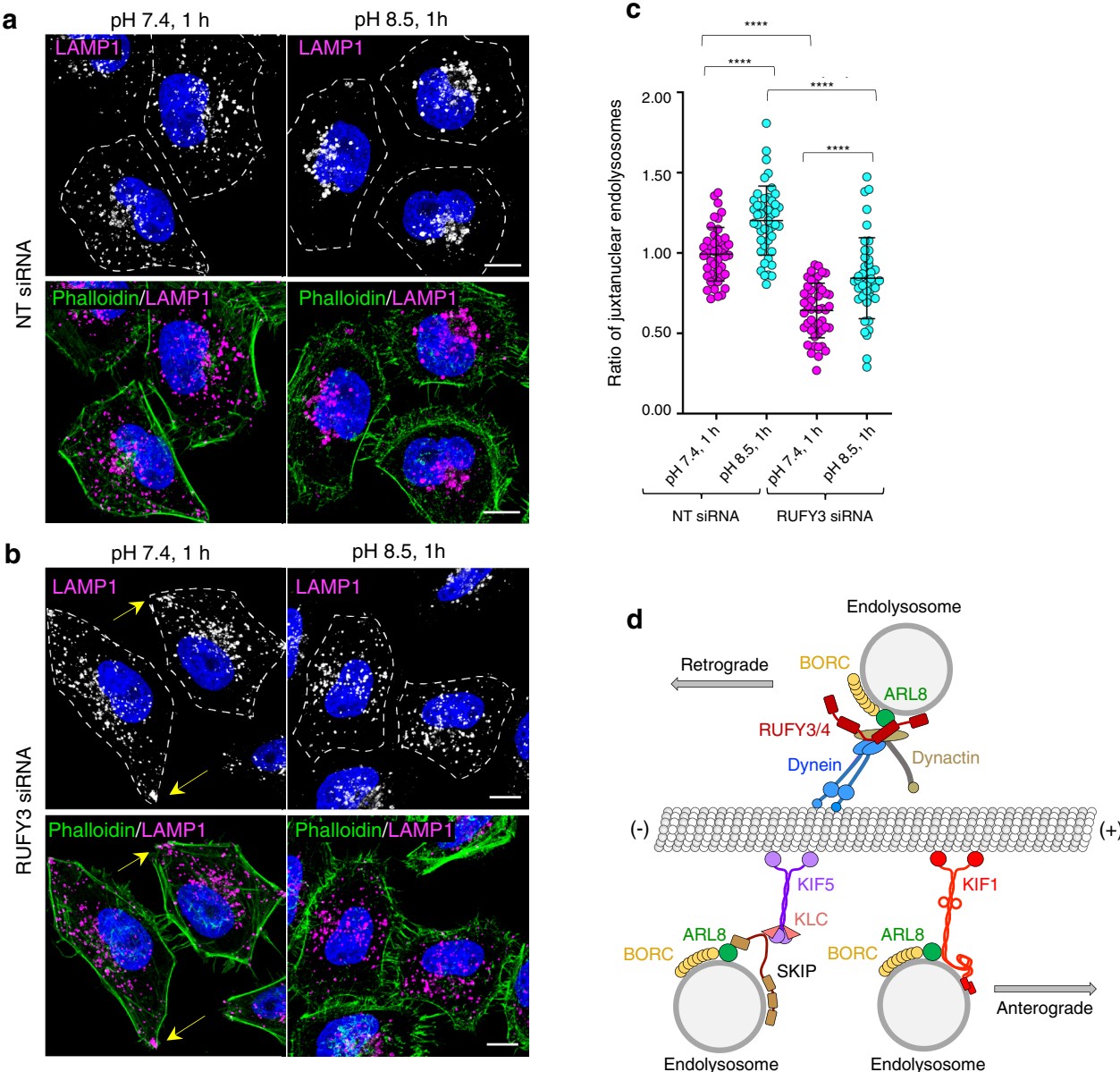

**Fig. 9 Requirement of RUFY3 for juxtanuclear clustering of endolysosomes upon cytoplasmic alkalinization, and schematic representation of the roles of ARL8 in retrograde and anterograde endolysosome transport. a, b** HeLa cells were treated with non-targeting (**a**) or RUFY3 siRNA (**b**) for 96 h, and left untreated (control) or incubated for 1 h at 37 °C in regular culture medium adjusted to pH 8.5. Cells were then fixed, permeabilized and immunostained with antibody to endogenous LAMP1 (grayscale and magenta) and Alexa Fluor 546-conjugated phalloidin (green). Nuclei were stained with DAPI (blue). Arrows point to accumulation of endolysosomes at cell tips caused by RUFY3 depletion. Scale bars: 10 μm. **c** Quantification of the ratio of juxtanuclear LAMP1 to total LAMP1 by shell analysis. The graph shows the individual data points and the mean ± SD from one experiment (panels (**a**) and (**b**)). Data were normalized to untreated cells in regular culture medium. Statistical significance was calculated by two-way ANOVA with multiple comparison between groups using Tukey's test. ****$p < 0.0001$. **d** Schematic representation of the role of ARL8 in regulating both retrograde and anterograde of endolysosomes through interactions with different effectors. BORC promotes recruitment of ARL8 to endolysosomes[7]. In turn, ARL8 recruits RUFY3 or RUFY4, which promotes coupling to dynein-dynactin. Interaction of ARL8 with RUFY3 is mediated by the CC2 domain. The domain of RUFY4 that interacts with ARL8 was not identified. These interactions drive transport of endolysosomes from the plus to the minus end of microtubules (i.e., retrograde transport) (this study). Additional RUFY3 or RUFY4 interactors not represented here may also participate in this coupling[91]. Alternatively, ARL8 recruits kinesin-1 (KIF5$_2$-KLC$_2$) via SKIP, or kinesin-3 (KIF1) directly, driving endolysosome transport from the minus to the plus end of microtubules (i.e., anterograde transport)[7,11,17,20].

antibodies diluted in blocking buffer for 30 min at 37 °C, washed 3 times with PBS, incubated with secondary antibodies diluted in blocking buffer for 30 min at 37 °C, washed twice with PBS and once with distilled water, and mounted on slides using Fluoromount-G with DAPI (Cat# 0100-20, Electron Microscopy Sciences). Alexa Fluor 546-phalloidin was added for 15 min at room temperature, after the secondary antibody was removed and the coverslip was washed 3 times in PBS.

**Image acquisition**. Images were acquired on a Zeiss LSM780 or Zeiss LSM880 inverted confocal laser scanning microscope fitted with a Plan-Apochromat 63X, 1.4 numerical aperture (NA) objective (Carl Zeiss). Live-cell imaging was performed in a controlled chamber (37 °C and 5% CO$_2$). Z-stacks were obtained, and maximal intensity projections were generated. Microcopy images were acquired with Zeiss ZEN Black software: Zen 2012 SPF FP3 release version 14.0.22.201, and

Zen 2.3 SP1 FP3 release version 14.0.25.201. Images were further processed in ImageJ[102] or FiJi vl.52p (NIH, Bethesda, MD).

**Endolysosome positioning measurements.** To quantify endolysosome positioning (Figs. 3h, 4d, 5b, e, f, 9c and Supplementary Fig. 3c, d), we used a "shell analysis"[69]. Briefly, z-stack confocal fluorescence micrographs of cells were flattened and a threshold was applied to eliminate background. Cells with a relatively centered nucleus and uniform shape were selected for the analysis, as narrow, elongated cells, could not be accurately analyzed. These criteria were pre-defined and applied to all conditions. Cells meeting these criteria were manually traced in ImageJ/Fiji using either cytosolic GFP signal or phalloidin-stained cortical actin for visualization. The total area corresponding to RUFY3 or LAMP1 signal in the cell was measured. Then, the cell outlines were consecutively reduced in size by a fixed length a total of 5 times, and the RUFY3 or LAMP1 area scored each time. Such an approach resulted in 5 shells within the cell, with shell 1 covering the cell vertices and shell 5 the perinuclear region. The RUFY3 or LAMP1 signal area within shell 5 was calculated as a percentage of total RUFY3 or LAMP1 area to give the percent perinuclear signal. The RUFY3 or LAMP1 signal area within shell 1 was calculated as a percentage of total RUFY3 or LAMP1 area to give the percent peripheral signal.

**Co-localization analysis.** Co-localization analysis (Figs. 3c, 4b) was done using the Pearson-Spearman correlation (PSC) plug-in for ImageJ/Fiji[102]. In the scatter plots of co-localization we report the Pearson correlation coefficient[103], representing the relationship of the signal intensity from green (transgenic GFP or GFP-RUFY constructs) and red (endogenously labeled LAMP1 or transgenic ARL8B-mCherry) channels of analyzed images. This value can range from −1 to +1, where 0 indicates no relationship and −1 and +1 indicates strong negative and positive correlation, respectively. In a given image, individual cells were masked prior to analysis using the selection brush tool as described[103] to determine the Pearson correlation coefficient per cell of GFP and LAMP1 or ARL8B-mCherry signals. A threshold level of 10 was set, under which pixel values were considered noise and not included in the statistical analysis. Three experimental replicates were done. The mean Pearson correlation coefficient per cell from each replicate experiment was plotted, and statistical significance between conditions was determined using one-way ANOVA with multiple comparisons to the GFP control or two-tailed unpaired Student's $t$ test ($n = 3$).

**Quantification of RUFY3 membrane association.** To compare membrane association of various RUFY3 deletion constructs from live-cell micrographs (Fig. 3f, g), GFP fluorescence before and after masking diffuse cytoplasmic signal was measured. Briefly, z-stack confocal fluorescence micrographs of live cells weakly expressing RUFY3-GFP plasmids were flattened and both total cellular fluorescence as well as fluorescence remaining after diffuse cytoplasmic signal was masked with a manual threshold was measured for each cell. Bright, non-diffuse, signal remaining after masking was attributed to accumulation of RUFY3-GFP constructs on endolysosomal membranes. This membrane-associated signal was plotted as a percent of total cellular GFP fluorescence for each cell and compared between deletion constructs.

**Manual scoring of microcopy experiments.** Scoring of cells in which RUFY proteins localized to mitochondria (Figs. 1f, 2e and Supplementary Fig. 1d, f) was done by visually scoring cells based on the RUFY-GFP signal. A minimum of 300 cells per condition from a minimum of three independent experiments were scored. Scoring of peroxisome distribution (Fig. 8e) was done by visually scoring cells based on the peroxisome phenotype that was detected by the RFP signal of the PEX3$_{1-42}$-FKBP-RFP plasmid for juxtanuclear, partially juxtanuclear and dispersed peroxisomes. A minimum of 300 cells per condition from a total of three independent experiments were scored, except for the BICD2 construct in the NT siRNA + Rapalog condition in which 200 cells from two experiments were used for the analysis.

**Co-immunoprecipitation.** $2.5 \times 10^6$ HEK293T cells were plated on 10-cm dishes and transfected the following day. Following transfection, cells were scraped and washed 3 times in cold PBS for 5 min at 4 °C with a $500 \times g$ spin between washes. Cell pellets were resuspended in 1 mL cold lysis buffer and incubated for 30 min at 4 °C with gentle rotation. In Fig. 2b, the lysis buffer composition was 25 mM Tris-HCl pH 7.4, 0.15 M NaCl, 1 mM EDTA, 1% NP-40 (Cat# 011332473001, Roche), 5% glycerol, supplemented with complete EDTA-free protease inhibitor tablet (Cat# 1836170, Roche). Following lysis, the soluble fraction was separated by centrifugation for 10 min at 4 °C, $17,000 \times g$. Lysates were incubated with 20 µl anti-FLAG magnetic agarose suspension (Cat# A36797, Thermo Fisher) overnight at 4 °C with gentle rotation. Following incubation, beads were washed 3 times in 1 mL of 25 mM Tris-HCl pH 7.5, 150 mM NaCl, 0.05% Tween-20 for 5 min at 4 °C, with a $500 \times g$ spin between washes. Washed beads were eluted by addition of Laemmli sample buffer and heating for 10 min at 99 °C.

In Fig. 7a, lysis buffer composition was 25 mM HEPES pH 7.4, 1 mM DTT, 0.2% NP-40, 0.5 mM Mg-ATP, 1 mM EGTA, 10% glycerol, 2 mM magnesium acetate, 50 mM potassium acetate, supplemented with complete EDTA-free

protease inhibitor tablet. Lysates were incubated with 30 µl magnetic GFP-Trap (homemade, detailed below) at 4 °C, 2 h with gentle rotation. Following incubation, cells were washed with lysis buffer without complete EDTA-free protease inhibitor tablet.

**Real-time qRT-PCR.** To determine KD efficiency, we used quantitative reverse transcription PCR (qRT-PCR). Briefly, total RNA was extracted from cells treated with non-targeting siRNA (siNT) or siRNA targeting RUFY3 or RUFY4, using the RNeasy Mini Kit (Cat# 74106, Qiagen) according to the manufacturer's instructions. Complementary DNA was generated by reverse transcription using the Superscript VILO cDNA Synthesis Kit (Cat# 11754050, Thermo Fisher), using 50 ng of the extracted mRNA as template. The cDNA was diluted 1:100 in PCR-grade water and used as template for qPCR with TaqMan® Gene Expression assays (Thermo Fisher) targeting either human RUFY3 (Cat# 4448892, Hs01127885_m1), RUFY4 (Cat# 4448892, Hs01651015_m1) or the housekeeping gene ACTB (Cat# 4448489, Hs01060665_g1) in the TaqMan Fast Advanced Master Mix (Cat# 4444557, Thermo Fisher). qPCR was performed on the AriaMx Real-Time PCR system using AriaMx software version 1.3 (Agilent Technologies).

**Preparation of rat hippocampal neurons.** Our study protocol for the preparation of rat hippocampal neurons followed all NIH ethical regulations. All research conducted in this study has been approved by the Intramural Research Program of NICHD. The animal procedure of rats was conducted following the NIH Guide for the Care and Use of Laboratory Animals, under protocols #19-011 and approved by the Animal Care and Use Committee of NICHD. The rats used in this study had normal health status. They were naive, i.e., they were not treated with any drugs and free from any pathogens. Pregnant albino rats were obtained to the animal facility on day 17 of gestation from ENVIGO RMS INC (Stock No. Sprague Dawley® SD®) and were maintained under a 12-hour dark and light cycle for 24-hours. Animals were sacrificed by carbon dioxide inhalation followed by decapitation prior to embryo extraction. Hippocampal neuron isolation and analysis were from pooled embryos of the same litter. Neurons were cultured and randomly allocated for different experiments like staining, live imaging with transfection of plasmid DNA.

Rat hippocampal neurons were isolated as previously described[104]. Briefly, E18 rat embryos were harvested and euthanized. The brains were isolated in Hank's medium, and hippocampi were dissociated mechanically with a narrow-mouth glass pipette followed by trypsinization with 0.25% trypsin (Cat# 1509046, Gibco) for 15 min at 37 °C. Cells were plated on 18-mm microscopic glass coverslips coated with polylysine (Cat# 11243217001, Roche) and laminin (5 µg/mL) (Cat# P2636, Millipore-Sigma) in DMEM with 4.5 g/L glucose, 25 mM HEPES, 10% heat-inactivated horse serum (Cat# 26050-088, Gibco), 100 U/mL penicillin and 100 µg/mL streptomycin. Three hours post plating, the medium was replaced with Neurobasal medium (Cat# 21103-049, Gibco), supplemented with 1X B27 (Cat# 17504044, Thermo Scientific), Glutamax (Cat# 35050-61, Life Technologies), and 100 U/mL penicillin-streptomycin (Cat# 15140148, Gibco) and placed at 37 °C and 5% $CO_2$.

**Transfection and immunofluorescence microscopy of neurons.** Rat hippocampal neurons were transfected at day-in-vitro 4 (DIV4) using 1.2 µL Lipofectamine 2000 mixed in 200 µL of Opti-MEM with 1–2 µg plasmid DNA per 18-mm cover glass with 800 µL Neurobasal medium for 1 h at 37 °C. After 1 h, Lipofectamine 2000 was washed with Neurobasal medium and the cells were kept in fresh, complete Neurobasal medium for 24 h. For immunofluorescence microscopy, neurons were fixed with 4% PFA in PBS supplemented with 4% sucrose, 0.1 mM $CaCl_2$ and 1 mM $MgCl_2$ (PBS-CM) for 20 min. Cells were permeabilized with 0.2% v/v Triton X-100 for 15 min at room temperature. After that, cells were incubated with 0.2% gelatin in PBS-CM for 30 min. Primary and secondary antibodies were prepared in blocking solution and incubated for 30 min each at 37 °C. Cells were mounted with Fluoromount G (Electron Microscopy Sciences). Images were taken in a Zeiss LSM780 confocal microscope using a Plan Apochromat 63x objective (N.A. 1.40).

**Live imaging of neurons.** To analyze endolysosome movement, neurons were co-transfected at DIV4 with plasmids encoding RUFY3-GFP or RUFY4-GFP along with LAMP1-RFP or ARLB-mCherry and imaged 24 h post-transfection. In live neurons, axons were identified by labeling the axon initial segment (AIS) protein neurofascin[104] (CF640R Mix-n-Stain antibody labeling kit, Cat # 92258, Biotium Science). Videos were recorded at 200 milliseconds for individual channels without any delay for a total of 5 min. Live-cell imaging was performed on a spinning-disk Eclipse Ti Microscope System (Nikon) equipped with a humidified environmental chamber maintained at 37 °C and 5% $CO_2$. Images were acquired with NIS-Elements AR microscope imaging software using a high-speed EMCCD camera (iXon Life 897, Andor).

To check the acidity of neuronal endolysosomes, neurons transfected with plasmids encoding GFP (control), RUFY3-GFP or RUFY4-GFP were labeled for 15 min with 1 µl of the cell membrane permeable dye LysoTracker red (LysoTracker red DND-99, Cat#L7528, Thermo Scientific) in 1 ml complete

Neurobasal medium at 37 °C. AIS staining and live imaging were done as mentioned above.

Axonal kymographs were generated using Fiji software with a segmented line tool of one-pixel thickness along a 50 μm segment of the axon just distal to the AIS, followed by stack re-slicing projection. In the kymographs, particles moving in anterograde and retrograde directions form lines with negative and positive slopes, respectively. Stalled particles were not included in any of the analyses. To measure the processivity of the moving endolysosomes, velocity and run length were calculated from the generated kymographs. Velocity of moving tracks was analyzed by measuring the tracks (μm) traveled by given time (s) in 50 μm long kymographs. Run length was analyzed by measuring the length of the tracks present in the 50 μm axon length kymographs using Fiji ROI manager. Co-localization of ARL8B-mCherry and LysoTracker red labeled endolysosomes with RUFY3-GFP or RUFY4-GFP was analyzed from total moving tracks present in the kymographs. The directionality (anterograde and retrograde) of the co-localized tracks was also quantified from the kymographs.

**Expression and purification of recombinant proteins**. BL21-CodonPlus (DE3) RP *E. coli* cells (Cat# 230255, Agilent Technologies) expressing target proteins were grown in 1 L Terrific Broth supplemented with 34 μg/mL chloramphenicol (C-6378, Millipore-Sigma) and 100 μg/mL ampicillin (Cat# A1066, Millipore-Sigma) for GST-plasmids or 30 μg/mL kanamycin (Cat# K1377, Millipore-Sigma) for 6His-StrepII-sfGFP plasmids. Cultures were grown for 6–8 h at 37 °C, with 200 rpm rotation, induced with 1 mM IPTG (isopropyl β-D-1-thiogalactopyranoside) (Cat# I2481, GoldBio) and incubated overnight at 16-18 °C, 200 rpm. Bacterial cultures were pelleted by centrifugation for 20 min at 4 °C, 6000 × g. and pellets were resuspended in lysis buffer supplemented with lysozyme (Cat# VWRV0663, VWR), DNase I (Cat# LS002139, Worthington Biochemical Corporation) and complete EDTA-free protease inhibitor tablet (Cat# 1836170, Roche) (specific buffers used are listed below for each protein purified in this study). Following sonication and centrifugation for 30-45 min at 4 °C, 35,267 × g, cleared lysates were incubated with glutathione-Sepharose 4B (Cat# 17-0756-05, Cytvia) (for GST- tagged proteins) or cOmplete-His-Tag Purification Resin (Cat# 5893682001, Roche) (for 6His-StrepII-sfGFP-tagged proteins) for 1–2 h at 4 °C, with gentle end-to-end rotation.

For purification of GST-ARL8B-Q75L and -T34N, lysis buffer was 50 mM Tris-HCl pH 8.0, 150 mM NaCl, 8 mM MgCl₂, 5% glycerol and 5 mM β-mercaptoethanol (Cat# M6250, Millipore-Sigma) supplemented with 100 μM GDP (Cat# G7127, Millipore-Sigma) (for GST-ARL8B-T34N) or 100 μM GTPγS (Cat# G8634, Millipore-Sigma) (for GST-ARL8B-Q75L). Bound glutathione-Sepharose was washed in buffer containing 50 mM Tris-HCl pH 8.0, 150 mM NaCl, 8 mM MgCl₂, 5% glycerol and 5 mM β-mercaptoethanol (Cat# M6250, Millipore-Sigma) supplemented with 100 μM GDP (G7127, Millipore-Sigma) (for GST-ARL8B-T34N) or 100 μM GTPγS (Cat# G8634, Millipore-Sigma) (for GST-ARL8B-Q75L).

GST-DLIC1 and GST-DLIC-CT were expressed in BL21(DE3) (Cat# C2527I, New England Biolabs) and lysis buffer was 50 mM Tris-HCl pH 8.0, 150 mM NaCl, 10% glycerol, and 1 mM DTT (Cat# DTT-RO, Roche). Following binding to glutathione Sepharose and washes, bound protein was eluted from the glutathione-Sepharose with buffer containing 50 mM Tris-HCl pH 8.0- and 10-mM L-glutathione. Eluant was further purified on HiLoad 16/60 Superdex 200 (Cat# 28-9893-35, Cytvia) in buffer containing 10 mM Tris-HCl pH 7.0, 50 mM NaCl, 2 mM MgCl₂ and 2 mM Tris(2-carboxyethyl)phosphine hydrochloride (TCEP) (Cat# C4706, Millipore-Sigma). Peak fractions were pooled together, aliquoted, flash-frozen in liquid nitrogen and stored at −80 °C.

For 6His-StrepII-sfGFP-RUFY3, 6His-StrepII-sfGFP-BICD₂₂₅₋₄₀₀ and 6His-StrepII-sfGFP purification, lysis buffer was 50 mM Tris-HCl pH 8.0, 300 mM NaCl, 5% glycerol and 1 mM DTT (Cat# 10708984001, Millipore-Sigma). Bound proteins on cOmplete-His-Tag Purification Resin (Cat# 5893682001, Roche) were washed in buffer containing 50 mM Tris-HCl pH 8.0, 300 mM NaCl, 5% glycerol and eluted in buffer composed of 50 mM Tris-HCl pH 8.0, 300 mM NaCl, 5% glycerol with 1 mM DTT and 90 mM imidazole. Proteins were further purified on Superose 6 Increase 10/300 column (Cat# 29-0915-96, Cytvia) in 50 mM Tris-HCl pH 8.0, 300 mM NaCl, 5% glycerol and 1 mM DTT. Peak fractions were pooled, aliquoted, flash-frozen in liquid nitrogen and stored at −80 °C.

**Preparation of GFP-nanobody conjugated agarose**. Homemade GFP-Trap beads were generated by first purifying the GFP nanobody, then coupling it to N-hydroxysuccinimide (NHS) beads (Cat# GE28-9513-80, Millipore-Sigma). *E.coli* BL21 (DE3) was transformed with pOPINE-GFPnanobody plasmid[99] (a gift from Brett Collins, Addgene #49172). GFP nanobody was expressed as described above with buffer composed of 50 mM Tris-HCl, 300 mM NaCl, 5% glycerol and 1 mM DTT that was supplemented with DNAse I, lysozyme and complete EDTA-free tablet. cOmplete His-Tag purification resin was prepared by washing 5 mL of resin with cold PBS. The cleared lysate was incubated in batch mode with the cOmplete His-Tag purification resin for 30 min at 4 °C, with end-to-end rotation. The lysate was removed, the resin washed with 600 mL cold PBS and the proteins eluted in PBS supplemented with 500 mM imidazole. The elution was conducted 4 times for a total elution volume of 20 mL. Eluant was dialyzed in 4 L PBS supplemented with 150 mM NaCl overnight at 4 °C. The nanobody was additionally purified by gel filtration on a Superdex 200 Increase 300/10 column (Cat# 28-9909-44, Cytvia) in

25 mM HEPES pH 7.4, 150 mM NaCl. Peak fractions were pooled and purified nanobody at 1.8 mg/ml concentration was aliquoted, flash frozen and stored at -80 °C while 1 mg was used to prepare GFP-Trap beads.

Coupling of the nanobody to NHS Mag Sepharose (Cat# GE28-9513-80, Millipore-Sigma) was conducted according to the supplier's specifications. Briefly, one 500 μL tube of NHS Mag Sepharose was placed on a magnetic rack and the storage solution was removed. The beads were equilibrated by resuspending them in 500 μL ice-cold 1 M HCl and removing the liquid. The 1 mg of purified nanobody, diluted to 1 mL in 0.2 M NaHCO₃, 0.5 M NaCl, pH 8.3, was added to the beads and allowed to mix end-over-end at room temperature for 20 min. The nanobody solution was then removed and residual active groups were blocked by sequential washes in 50 mM Tris-HCl, 1 M NaCl, pH 8 (Buffer A) and 50 mM glycine-HCl, 1 M NaCl, pH 3.0 (Buffer B). The washes were as follows: 500 μL Buffer A, 500 μL Buffer B, 500 μL Buffer A, mixed end-over-end at room temperature for 15 min. The buffer was removed. The beads were sequentially washed in 500 μL Buffer B, 500 μL Buffer A and 500 μL Buffer B. The beads were resuspended in 500 μL of 50 mM Tris-HCl pH 7.4 containing 20% ethanol and stored at 4 °C.

**Pull downs**. HEK293T cells expressing the indicated RUFY constructs and controls were scraped from 10-cm plates and washed 3 times in 1 mL cold PBS followed by centrifugation for 5 min at 4 °C, 500 × g. Pellets were resuspended in 1 mL buffer containing 25 mM Tris-HCl pH 7.4, 150 mM NaCl, 1 mM EDTA, 1% NP-40 (Cat# 011332473001, Roche) and 5% glycerol, supplemented with EDTA-free protease inhibitor tablet (Cat# 1836170, Roche) and 500 μM GDP (Cat# G7127, Millipore-Sigma) (for GST-ARL8B-T34N) or 500 μM GTPγS (Cat# G8634, Millipore-Sigma) (for GST-ARL8B-Q75L), 1 mM DL-dithiothreitol (DTT) (Cat# 10708984001, Millipore-Sigma) and 8 mM MgCl₂,and incubated for 30 min at 4 °C with gentle rotation. Lysates were centrifuged for 10 min at 4 °C, 17,000 × g and incubated with 20 μl glutathione-Sepharose loaded with GST-ARL8B-Q75L or GST-ARL8B-T34N for 1 h at 4 °C with gentle rotation (preparation of GST-ARL8B-Q75L and -T34N is described above). Bound material was separated by centrifugation for 5 min at 4 °C, 500 × g, and washed 3 times with 1 mL buffer containing 50 mM Tris-HCl pH 8.0, 150 mM NaCl, 8 mM MgCl₂, 5% glycerol and 5 mM β-mercaptoethanol (Cat# M6250, Millipore-Sigma) supplemented with 100 μM GDP (Cat# G7127, Millipore-Sigma) (for GST-ARL8B-T34N) or GTPγS (G8634, Millipore-Sigma) (for GST-ARL8B-Q75L) (Cat# 10708984001, Millipore-Sigma). Samples were eluted with Laemmli sample buffer for 10 min at 99 °C.

For the pull down with GST-DLIC and GST-DLIC-CT, 20 μg protein was incubated with 20 μl glutathione-Sepharose. Loaded GST-beads were incubated with 5 μg purified 6His-StrepII-sfGFP-RUFY3 for 1 h at 4 °C with gentle rotation and processed as above.

**Endogenous dynein pulldown**. We used a published protocol[76,105] with modifications. HEK293T cells from fifteen 15-cm plates were scraped and washed 3 times in cold PBS for 5 min at 4 °C, with 500 × g centrifugations between washes. Cells were lysed in 15 ml buffer composed of 25 mM HEPES pH 7.4, 5 mM DTT, 0.2% NP40, 1 mM Mg-ATP, 1 mM EGTA, 10% glycerol, 2 mM magnesium acetate, 50 mM potassium acetate, supplemented with complete EDTA-free protease inhibitor tablet for 1 h at 4 °C with gentle rotation. The supernatant was separated by centrifugation for 30 min at 4 °C and 120,000 × g (TLA45). 3.5 mL of the cleared HEK293T lysate was mixed with 100 μl Strep-Tactin Sepharose resin (Cat# 2-1201-010, IBA) and also 40 μg purified 6His-StrepII-sfGFP-RUFY3, 6His-StrepII-sfGFP-BICD2 and 6His-StrepII-sfGFP and incubated overnight at 4 °C with gentle rotation. Following incubation, beads were washed 5 times in 2 mL buffer for 3 min at 4 °C with 500 × g spins between washes. Samples were further eluted with 50 μl 4X Laemmli sample buffer, 10 min at 99 °C.

**Statistical calculations**. All statistical tests were performed on three independent experiments, except in Fig. 9c, and Supplementary Fig. 3c, d where the number of cells was used for statistics. Data are presented as SuperPlots[106]. These plots show individual data points in small circles, the mean from each experiment in big circles (both color coded per experiment), and the mean ± SD from the means of each experiment. We used one-way ANOVA when multiple groups were compared, and two-tailed unpaired Student's *t* test when two groups were compared. Data in Figs. 4d, 5b, e, f, and 9c, were normalized to the control. This was done to account for experiment-to-experiment variability (the trends within each experiment were always consistent). Statistical tests and graphs were made with Prism v9 GraphPad (San Diego, CA, USA).

**Reporting summary**. Further information on research design is available in the Nature Research Reporting Summary linked to this article.

## Data availability

The mass spectrometry raw data generated in this study have been deposited in the MassIVE database as MSV000087741 [https://massive.ucsd.edu/ProteoSAFe/dataset.jsp?task=dc443eda7bb04ba196f3430b9f346b1a]. The processed mass spectrometry data are available as Supplementary Data 1. Microscopy data that support our findings are available upon request from the corresponding author. Source data are provided with this paper.

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

## Acknowledgements

The authors thank for Xiaolin Zhu and Boma Fubara excellent technical assistance, Anna Akhmanova, Cecilia Bucci, Brett Collins, Wade Harper, Steve Jackson, Walter Mothes, Sean Munro, Kyle Roux, and Ron Vale for kind gifts of reagents, and other members of the Bonifacino lab for helpful discussions. This work was supported by the Intramural Program of NICHD, NIH (project # ZIA HD001607 to JSB).

## Author contributions

T.K.K and J.S.B conceived the project. T.K.K designed and conducted most of the experiments. A.S. contributed reagents, conducted and analyzed experiments with shell analysis and qRT-PCR. S.G. conducted and analyzed experiments in neurons. C.W. contributed to live-cell imaging and data quantification. R.J helped with experiments on endolysosome repositioning. Y.L conducted mass spectrometry. T.K.K and J.S.B wrote the manuscript with input from all the authors.

## Funding

## Competing interests

The authors declare no competing interests.
