## [Peer Review File · Nature Communications]

RUFY3 and RUFY4 are ARL8 effectors that promote coupling of endolysosomes to dynein-dynactinREVIEWER COMMENTS

Reviewer #1 (Remarks to the Author):

In this study, Keren-Kaplan et al identify 2 novel effectors of the small GTPase Arl8, namely RUFY3 and RUFY4. They use proximity labeling coupled with mitochondrial targeting (Mito ID) to identify these ARL8 interactors. Using imaging and biochemical approaches, they demonstrate that RUFY3 and 4 localize to LAMP1 and ARL8 positive vesicles and drive perinuclear clustering of these organelles. They also show using in vitro experiments that RUFY3 and 4 interact with dynein-dynactin complex. Using deletion constructs they identify the CC2 domain on RUFY3 and 4 to be responsible for ARL8 binding and driving the perinuclear clustering. RUFY3 also appears to localize to a subset of axonal LAMP1-positive organelles, in culture primary neurons. Lastly, RUFY 3 / 4 appears to be required for the juxtannuclear movement of lysosomes upon serum starvation.

The study is interesting and novel in that it identifies new effectors of ARL8 GTPase and links this GTPase to retrograde transport for the first time. Through well-designed experiments, striking images and quantitative data, the authors convey most of the points mentioned above very clearly. There are some minor issues that should be addressed prior to publication:

1. Role of the different domains (CC2 and FYVE in particular):

a) In figure 3d, it appears that RUFY3 lacking RUN or CC1 domains, while still localize to lysosomes/vesicles, are much less 'tightly' clustered to perinuclear region. This could suggest a role for these domains in supporting the retrograde movement or in interaction with retrograde transport machinery? It would be helpful to see a quantification of the strength of this clustering.

b) There seems substantial recruitment of RUFY3 to vesicles even in absence of CC2 (Fig 3d) although the perinuclear clustering is clearly reduced, while the version of protein lacking CC2 and FYVE domains is largely cytosolic. Is it more accurate to so both are involved in recruitment to vesicles? Also, can the authors explain why this looks different from the mito -experiments?

2. RUFY3/4 function in neurons:

a) Kymographs in Fig 5e seem to suggest that RUFY4 is on both anterogradely and retrogradely moving LAMP1 vesicles while RUFY3 is largely on retrogradely moving ones. This looks a bit different from the quantification in 5h where both RUFY3/4 look to be more on retrogradely moving vesicles.

b) Likewise, while both seem to increase relative proportion of retrogradely moving LAMP1 vesicles, there is a strong reduction of total number of motile LAMP1 axonal vesicles. This data is very interesting overall but concluding that they increase retrograde transport seems a bit simplistic. Could RUFY3/4 affect Arl8 interaction/binding with SKIP and /or Kinesin? Likewise, are there changes to pauses in

LAMP1 vesicle motion, processivity, velocity? Further dissection of how RUFY3/4 over expression alters the motility will shed important information on the transport properties of these organelles.

c) As the authors point out in their discussion RUFY3/4 localizes to a subset of axonal LAMP1-positive organelles. Given the interest in these organelles, their maturation and transport in the neuronal cell biology field, it would help to further define this sub-population: are there Rab7 positive? Are they acidic (there is a strong correlation between retrogradely moving LAMP1 vesicles in axons and their acidic nature). These experiments will help determine more clearly, the nature of these RUFY3-positive endo-lysosomes.

d) While the authors clearly demonstrated a reduction of LAMP1 tracks, based on over expression, from their live imaging experiments (does this include stationary vesicles?), it would be better to examine the number of endogenous LAMP1 vesicles (per unit length of axon), to conclude that there are reduced numbers of lysosomes in axons.

Other minor points:

In Fig 7 e, it will be good to include Raplog in the graph's X axis as it is a bit confusing without that- it appears as if the +/- are for the siRNA.

Reviewer #2 (Remarks to the Author):

In this study, the authors reveal a new retrograde transport pathway for late endosomes/lysosomes mediated by a small GTPase Arl8b, which has thus far only been appreciated for its role in anterograde transport of the same organelles. Specifically, Keren-Kaplan et al identify RUFY3 and 4 as the first minus-end-directed transport effectors for Arl8b. The authors show that RUFY3 interacts with Arl8b using its CC2 domain, thereby allowing endosomes to be transported towards the perinuclear area in a dynein/dynactin dependent manner. The manuscript is well structured and easy to read. Most of the conclusions drawn are substantiated by the experiments presented, and the data appears reproducible and of high quality. It is important to point out however that the novelty of the findings is threatened by a recent preprint by Kumar and colleagues (DOI: 10.21203/rs.3.rs-345822/v1). Irrespective of this, several important issues would need to be addressed prior to publication:

Major points

1. My biggest concern is the implicit assumption by the authors that all retrograde transport mediated by RUFY3/4 results downstream of Arl8b. This approach disregards the possibility that Rab7 may (also) be directing these actions. It is well established that a subset of late endosomes/lysosomes contains both Arl8 and Rab7, and several other Arl8 effectors, including SKIP and the HOPS complex are shared by

these GTPases. The authors should address this point, for example by examining whether Rab7 (QL versus TN) can interact with RUFY3/4.

2. The authors mention that they find that binding of Arl8 to RUFY involves the CC2 domain of RUFY3. Although the truncation data indicates necessity, the authors should include evaluation of the CC2 domain (or combined with FYVE) to demonstrate sufficiency with respect to the recruitment to Arl8b.

3. It would have been nice to see that engagement of RUFY3/4 with peripheral Arl8b-positive endosomes drives their transport into the juxtannuclear region in live cells.

Minor points

4. In figure 2g, dCC2-FLAG and dCC2dFYVE-FLAG are not pulled down with GST-Arl8b-QL, however a band is visible for the condition where they used GST-Arl8b-TN. What could be the reason for this?

5. RUFY3 siRNAs need to be deconvoluted to show that multiple RUFY3 siRNA duplexes show similar effects.

6. Figure 4h: in the legend it is not mentioned what the yellow-colored arrows refer to.

7. Figure 7e is missing the annotation +/- Rapalog.

8. Figure 6c Annotation of GST-DLIC1 and GST-DLIC-CT is swapped.

9. Some of the clustering pictures in Fig4 are not convincing enough (compared to the quantification). For example, β RUN-GFP does not look as clustered as RUFY3-GFP, while the quantifications show similar average clustering.

Reviewer #3 (Remarks to the Author):

The manuscript by Keren-Kaplan et al. is a well written study that describes identification of novel ARL8 GTPase effectors, RUN- and FYVE-domain containing proteins, RUFY3 and RUFY4 that are important for retrograde lysosome movement in cells. The authors use a combination of cell biology and biochemical methods to show that RUFY3 and RUFY4, but not other members of the RUFY family, associate with ARL8-lysosomes. The authors also map regions in RUFY3 important for an interaction with ARL8. Furthermore, the authors provide evidence that the retrograde movement of ARL8-lysosomes is mediated by RUFY3 and RUFY4 association with the dynein complex, although direct evidence of binding to dynein is only shown for RUFY3. Together with previous studies from this group and others showing that ARL8 can associate with kinesin adaptors to move lysosomes in an anterograde direction, this work provides important insights into how ARL8 regulates movement of lysosomes in cells.

The experiments are described in sufficient detail to be replicated and statistical analysis appears appropriate. I think that this study will be of high interest to the field of cellular trafficking, and I recommend its publication in the journal of Nature Communications.

I have few minor suggestions described below:

1. In section titled: "ARL8B promotes recruitment of RUFY3 and RUFY4 to a juxtannuclear cluster of vesicles" authors conclude that ARL8 promotes the recruitment of RUFY3 and RUFY4 vesicles via the CC2 domain and that FYVE domain makes additional contributions to this recruitment. However, the contributions of RUFY4 domains are never tested. I would suggest clarifying this statement to focus only on domain contributions from RUFY3.
2. The different RUFY3-GFP truncation constructs shown in inserts in figure 4e look very cytoplasmic, which is not fully consistent with images shown in figure 3d. Is there an increase in cytoplasmic localization of these constructs when co-expressed with LAMP1? If so, this would be unexpected and should be addressed in the text. It would also be helpful if the authors included full size images in the main figure or in supplementary figure.
3. There is a typo in the legend for figure 1, panel f is labeled with an e.
4. Figure 2 panel a, please add description of what is being blotted in the input.
5. Figure 2 panel e has RUN-GFP included although there is no description of this construct anywhere in the text or figure legend. Similarly, there is RUN-FLAG included in panel f and g, but again there is no description of this construct. Please add description of these constructs.
6. In general, all labels for microscopy images are very small and hard to see. Could the authors please increase font size or rearrange labels to be above or below images, so it is easier to see them?
7. Please add reference for the source of ARL8A-B-KO cells.

Reviewer #4 (Remarks to the Author):

In this manuscript, the authors identified RUFY3.1, a previously uncharacterized longer isoform of RUFY3, and RUFY4 as novel ARL8 effectors by using the MitolD method. They then found that both RUFY3.1 and RUFY4 promote retrograde transport of lysosomes through physical interaction with the dynein-dynactin motor complex and that their activity is required for accumulation of lysosomes in the juxtannuclear area of cells. Overall, the experiments are well conducted and there is good inclusion of positive and negative controls to reveal the selectivity and specificity of the observations. Thus, the study makes a significant contribution to the mechanistic understanding of ARL8-mediated lysosomal positioning that plays important roles in a wide range of cellular processes.

Specific comments:

1. In Figure S1c, the authors showed that RUFY3.2, an originally described RUFY3 isoform, is present in the cytosol in HeLa cells, strongly suggesting that RUFY3.2 is not an ARL8 effector. In the present manuscript, however, it is not clear whether RUFY3.2 binds to ARL8. The authors should test their interaction biochemically.
2. Based on the results of Figure 3d, the authors concluded that in addition to the ARL8-interacting CC2 domain, the FYVE domain also contributes to the RUFY3.1 localization to cytoplasmic vesicles. Is RUFY3.1 localized to cytoplasmic vesicles via its FYVE domain independently of ARL8? The authors need to test the localization of RUFY3.1-deltaCC2 in ARL8A/B-KO cells.
3. In Figure 5i, expression of RUFY3-GFP or RUFY4-GFP promoted retrograde transport of lysosomes in axons but also “reduced the number of moving lysosomes”. Why? Please discuss the possible reason. Does expression of RUFY3-GFP or RUFY4-GFP also affect lysosome transport in “dendrites”?
4. Distinct Rab binding activities of the RUFY family members in the Discussion section is important (lines 397-404), but the information about the Rab33A binding activity of RUFY3.2 is missing.
5. Typos.
(line 40) unles; (line 357) RUFY3.3; (line 422) Once possibility; (line 683) Fig. 3c,4b should read Fig. 3b, 4b.
Figure 6c seems to be mis-labeled. The far right lane should be the GST-DLIC1 full-length protein.
Page numbers of several references are missing (e.g., lines 973,1026, 1064, 1082, 1096, and 1114).

Reviewer #5 (Remarks to the Author):

Keren-Kaplan et al. use an alternative BioID2 assay called MitoID to identify proximal proteins of ARL8A and ARL8B. This resulted in the identification of RUFY3 and RUFY4 proteins that bind to the GTP-bound form of ARL8. The authors further demonstrate the interaction with dynein dynein and link juxtannuclear redistribution of lysosomes under various conditions with the complex that they have described in this study.

For the proteomics part, the design of the experiment is not very clear in the current manuscript. While the authors briefly mention in the methods section that 3 biological replicates were used (with 3 '.raw' in the data files), this is not clear from the results part, nor from fig. S1A or the processed tables. It would be better to make this clear to the readers in the figures and the text. Accordingly, the data processing to obtain Figures 1B and 1C and Supplementary dataset 1 should be better explained in the manuscript. BioID and BioID2 data have a tendency to generate a lot of background (especially upon massive overexpression as is the case here) so careful data analysis and processing is essential and should be transparent. For example, how exactly was the abundance ratio obtained and how was this transformed to fit the 0-100 axis? Moreover, the proteomics data should be made available in a public repository so it becomes available to the community. It is not clear whether this is planned. Along this lines, it is also worthwhile to elaborate briefly on the benefits of MitolD over classical BioID (or BioID2) to better motivate the use of this interesting method to the readers.

Fig 4g. shows RT-qPCR analysis upon knockdown of the RUFY3. While I appreciate the other data supporting the link between RUFY3 and lysosomal localization, perturbation is important. The authors should clarify if this was a single siRNA or a pool of siRNAs. In the case of a single siRNA, it would be best to add another siRNA to eliminate the possibility of off-target effects. In the case of a pool, the pool should be deconvoluted to find (hopefully) at least 2 decent siRNAs that give the same phenotype.

Minor comments

1. Line 110. The originally described MitolD procedure attaches BioID and the mitochondrial targeting sequences C-terminally from the Ras family protein under investigation. Please provide some more explanation for switching the modules in this study.
2. Line 112. Provide a reference for the ARL8 mutants that lock the protein in the GDP and GTP-bound state.
3. Line 601: I assume that TECP is referring to the reducing chemical tris (2-carboxyethyl)phosphine so this should be TCEP.
4. Methods line 710-712. Transfection method and incubation time after transfection are not mentioned.
5. Methods line 718. "Following incubation, cells were washed" ? I think this should be "beads were washed"?
6. Fig 2b. On the bottom panel, many aspecific signals can be seen with very different intensities, please indicate the band sizes expected for each FLAG-tagged construct (for example in the figure legend).
7. Line 208, Fig 3c makes use of HeLa ARL8A-B KO cells, but these cells are not mentioned anywhere in the methods. Please describe their origin or how you established them, and provide evidence of successful KO if these cells have not been published elsewhere.
8. Fig 7e. Indicate which bars refer to rapalog-treated cells. Also check RUFY4 condition (NT and DHC siRNA)

Responses to Reviewers

We thank the reviewers for their insightful and positive comments (*italics*). Changes to the text are indicated in **red** in the attached manuscript file. Below is a point-by-point response to the reviewers' comments:

Reviewer #1

In this study, Keren-Kaplan et al identify 2 novel effectors of the small GTPase Arl8, namely RUFY3 and RUFY4. They use proximity labeling coupled with mitochondrial targeting (Mito ID) to identify these ARL8 interactors. Using imaging and biochemical approaches, they demonstrate that RUFY3 and 4 localize to LAMP1 and ARL8 positive vesicles and drive perinuclear clustering of these organelles. They also show using in vitro experiments that RUFY3 and 4 interact with dynein-dynactin complex. Using deletion constructs they identify the CC2 domain on RUFY3 and 4 to be responsible for ARL8 binding and driving the perinuclear clustering. RUFY3 also appears to localize to a subset of axonal LAMP1-positive organelles, in culture primary neurons. Lastly, RUFY 3 / 4 appears to be required for the juxtannuclear movement of lysosomes upon serum starvation.

The study is interesting and novel in that it identifies new effectors of ARL8 GTPase and links this GTPase to retrograde transport for the first time. Through well-designed experiments, striking images and quantitative data, the authors convey most of the points mentioned above very clearly. There are some minor issues that should be addressed prior to publication:

We thank this reviewer for the positive assessment of our manuscript. We addressed the issues mentioned by this reviewer as described below.

1. Role of the different domains (CC2 and FYVE in particular):

a) In figure 3d, it appears that RUFY3 lacking RUN or CC1 domains, while still localize to lysosomes/vesicles, are much less 'tightly' clustered to perinuclear region. This could suggest a role for these domains in supporting the retrograde movement or in interaction with retrograde transport machinery? It would be helpful to see a quantification of the strength of this clustering.

We have revised this figure to show transfected cells with more similar levels of expression, which allows for better comparison of the localization of RUFY3-GFP deletion constructs (new Fig. 3f). Overall, we did not see significant changes in juxtannuclear clustering of RUFY3 vesicles upon deletion of domains other than CC2 (more dispersed) (new Fig. 3h). Importantly, in the new version of the manuscript, we have additionally quantified the association of the RUFY3-GFP deletion constructs with vesicles (new Fig. 3g). We found that deletion of the RUN domain, alone or in combination with the CC1 domain, did not significantly alter association of RUFY3-GFP with vesicles (new Fig. 3g). Deletion of the CC1 domain alone decreased vesicle association, though to a much lesser extent than deletion of the CC2 or FYVE domains (Fig. 3g). These results clearly show that both the CC2 and FYVE domains promote membrane association of RUFY3 vesicles, and that the CC2 domain additionally promotes juxtannuclear clustering of the vesicles. Other domains appear to be less important for these processes.

b) There seems substantial recruitment of RUFY3 to vesicles even in absence of CC2 (Fig 3d) although

the perinuclear clustering is clearly reduced, while the version of protein lacking CC2 and FYVE domains is largely cytosolic. Is it more accurate to say both are involved in recruitment to vesicles?

The reviewer's observations are correct. The revised images in the new Fig. 3f better show the localization of different RUFY3-GFP deletion constructs with comparable levels of expression. The Δ CC2 construct exhibits partially decreased association with vesicles (new Fig. 3f, g), and these are less clustered in the juxtannuclear area (new Fig. 3f, h). The Δ FYVE construct is also less associated with vesicles (new Fig. 3f, g) and the remaining vesicles are tightly clustered in the juxtannuclear area (new Fig. 3f, h). The CC2 or FYVE domains alone are completely cytosolic (new Fig. 3f, g). We have also added a new Supplementary Fig. 2a showing that ARL8 KO decreases the clustering of full-length RUFY3 and abrogates the localization of the Δ FYVE construct to the tight juxtannuclear cluster. From these experiments, we conclude that both the CC2 and FYVE domains contribute to association with vesicles, and that the CC2 domain additionally contributes to juxtannuclear clustering. These findings are now described in more detail in the Results section. We thank the reviewer for raising this issue, which helped us provide a more precise description of the roles of the CC2 and FYVE domains.

Also, can the authors explain why this looks different from the mito -experiments?

The localizations of RUFY3/4 in Figs. 1 (mito-ID with the T34N ARL8 construct) and 3 (non-mito-ID) are not so different. If there is a subtle difference, it could be due to the fact that in the experiment in Fig. 1 cells were fixed, whereas in that in Fig. 3 cells were imaged live. The difference in the membrane recruitment of the CC2 domain alone in Figs. 2d and 3f is likely due to the overexpression of the mitochondrially targeted ARL8 in Fig. 2d. This is now explained in the text.

2. RUFY3/4 function in neurons:

a) Kymographs in Fig 5e seem to suggest that RUFY4 is on both anterogradely and retrogradely moving LAMP1 vesicles while RUFY3 is largely on retrogradely moving ones. This looks a bit different from the quantification in 5h where both RUFY3/4 look to be more on retrogradely moving vesicles.

We thank the reviewer for pointing out this discrepancy. It was due to a poor choice of the representative images. We have now replaced the kymograph analyses and chosen images that are more representative of the quantification (new Fig. 6e, f). Both the images and the quantification show that RUFY3 and RUFY4 are more abundant in retrogradely-moving vesicles.

b) Likewise, while both seem to increase relative proportion of retrogradely moving LAMP1 vesicles, there is a strong reduction of total number of motile LAMP1 axonal vesicles. This data is very interesting overall but concluding that they increase retrograde transport seems a bit simplistic. Could RUFY3/4 affect Arl8 interaction/binding with SKIP and/or Kinesin?

We agree with the reviewer that RUFY3/4 could compete with SKIP and kinesins for binding to ARL8, and we now mention this caveat in the Discussion. However, the fact that RUFY3 KD disperses endolysosomes, that RUFY3/4 interact with dynein-dynactin, and that they redistribute peroxisomes to the cell center in a dynein-dynactin-dependent manner

support their role as dynein-dynactin adaptors. We think increased retrograde transport is the most sensible interpretation for the effect of RUFY3/4 on axonal LAMP1 vesicles.

Likewise, are there changes to pauses in LAMP1 vesicle motion, processivity, velocity? Further dissection of how RUFY3/4 over expression alters the motility will shed important information on the transport properties of these organelles.

New quantifications shown in Figs. 6h, i demonstrate that RUFY3/4 have little or no effect on the velocity and run length of both anterograde and retrograde LAMP1 vesicles in the axon. The main effects are a decrease in the relative number of anterograde LAMP1 tracks and the total number of LAMP1 tracks in the axon. While we appreciate the opportunity to provide these additional data, we think that further analyses of axonal transport exceed the scope of this study and our possibilities at this time.

c) As the authors point out in their discussion RUFY3/4 localizes to a subset of axonal LAMP1-positive organelles. Given the interest in these organelles, their maturation and transport in the neuronal cell biology field, it would help to further define this sub-population: are there Rab7 positive? Are the acidic (there is a strong correlation between retrogradely moving LAMP1 vesicles in axons and their acidic nature). These experiments will help determine more clearly, the nature of these RUFY3-positive endolysosomes.

The identity of LAMP1 vesicles in the axon is currently a matter of debate, and we do not aspire to resolve this important but complex problem in this study. Nevertheless, we are happy to provide additional characterization of these vesicles in the new Supplementary Fig. 4. We find that RUFY3 and RUFY4 exhibit 100% co-localization with ARL8 in axonal vesicles (new Supplementary Fig. 4d-f). We also find that ~85% RUFY3 and ~40% RUFY4 are associated with LysoTracker-positive vesicles (new Supplementary Fig. 4h), and that RUFY3/4-LysoTracker-positive vesicles move mostly in the retrograde direction (new Supplementary Fig. 4i, j). These data indicate that RUFY3 is largely associated, and RUFY4 partially associated, with retrogradely-moving acidic endolysosomes. In addition, RUFY4 seems to associate with another population of non-acidic vesicles, the identity of which remains to be determined.

We did not examine the co-localization with RAB7 because this is a very complex issue that deserves its own, dedicated study, and because RUFY3 and RUFY4 do not interact with RAB7 (new data in Supplementary fig. 1e, f).

d) While the authors clearly demonstrated a reduction of LAMP1 tracks, based on over expression, from their live imaging experiments (does this include stationary vesicles?), it would be better to examine the number of endogenous LAMP1 vesicles (per unit length of axon), to conclude that there are reduced number of lysosomes in axons.

In the new Supplementary fig. 4a, b, we show that expression of RUFY3 or RUFY4 decreases the number of endogenous LAMTOR4 (a bona fide endolysosomal marker) puncta per unit length of axon. We used this particular marker because the antibody gives stronger, more specific staining. This finding is consistent with the reduction of not only moving LAMP1 tracks but also the overall number of endolysosomes in the axon.

Other minor points:

In Fig 7 e, it will be good to include Raplog in the graph's X axis as it is a bit confusing without that- it appears as if the +/- are for the siRNA.

We fixed this issue on the figure (now Fig. 8e).

Reviewer #2

In this study, the authors reveal a new retrograde transport pathway for late endosomes/lysosomes mediated by a small GTPase Arl8b, which has thus far only been appreciated for its role in anterograde transport of the same organelles. Specifically, Keren-Kaplan et al identify RUFY3 and 4 as the first minus-end-directed transport effectors for Arl8b. The authors show that RUFY3 interacts with Arl8b using its CC2 domain, thereby allowing endosomes to be transported towards the perinuclear area in a dynein/dynactin dependent manner. The manuscript is well structured and easy to read. Most of the conclusions drawn are substantiated by the experiments presented, and the data appears reproducible and of high quality. It is important to point out however that the novelty of the findings is threatened by a recent preprint by Kumar and colleagues (DOI: 10.21203/rs.3.rs-345822/v1). Irrespective of this, several important issues would need to be addressed prior to publication:

We thank this reviewer for his positive comments on our manuscript.

Major points

1. My biggest concern is the implicit assumption by the authors that all retrograde transport mediated by RUFY3/4 results downstream of Arl8b. This approach disregards the possibility that Rab7 may (also) be directing these actions. It is well established that a subset of late endosomes/lysosomes contains both Arl8 and Rab7, and several other Arl8 effectors, including SKIP and the HOPS complex are shared by these GTPases. The authors should address this point, for example by examining whether Rab7 (QL versus TN) can interact with RUFY3/4.

In the new Supplementary fig. 1e, f, we show that neither Q67L nor T22N forms of mito-RAB7A relocate RUFY3 and RUFY4 to mitochondria. This is in contrast to the mitochondrial relocation of a known RAB7A effector, RILP, by the Q67L construct (positive control). This demonstrates that RUFY3/4 are not RAB7 effectors. In light of these results, we did not further pursue a possible relationship of RUFY3/4 with RAB7A.

2. The authors mention that they find that binding of Arl8 to RUFY involves the CC2 domain of RUFY3. Although the truncation data indicates necessity, the authors should include evaluation of the CC2 domain (or combined with FYVE) to demonstrate sufficiency with respect to the recruitment to Arl8b.

We thank the reviewer for raising this important issue. We now provide new mito-targeting (new Fig. 2d, e) and pulldown (new Fig. 2h, i) data showing that the CC2 but not FYVE domain binds the active form of ARL8.

3. It would have been nice to see that engagement of RUFY3/4 with peripheral Arl8b-positive endosomes drives their transport into the juxtannuclear region in live cells.

While we acknowledge the importance of the suggested experiment, it would be difficult to perform in non-neuronal cells because RUFY3/4 causes strong clustering of endolysosomes in the juxtannuclear area. We think that our experiments in neurons provide evidence for an increase in the ratio of retrograde vs. anterograde LAMP1 vesicles (Fig. 6e, f), consistent with RUFY3/4 promoting retrograde transport.

Minor points

4. In figure 2g, *dCC2-FLAG* and *dCC2dFYVE-FLAG* are not pulled down with *GST-Arl8b-QL*, however a band is visible for the condition where they used *GST-Arl8b-TN*. What could be the reason for this?

This is a faint non-specific band that is seen in overexposed blots. For better appreciation of the relative intensity of this non-specific band vs. specific bands, in the new Fig. 2g we show a shorter exposure of the blots. In any case, both the long and short exposures are shown in the source data file.

5. *RUFY3* siRNAs need to be deconvoluted to show that multiple *RUFY3* siRNA duplexes show similar effects.

In the new Supplementary Fig. 3, we show that 3 of the 4 single siRNAs in the original SMARTpool effectively knock down *RUFY3*, and that the 3 effective siRNAs cause peripheral dispersal of endolysosomes.

6. Figure 4h: in the legend it is not mentioned what the yellow-colored arrows refer to.

We now indicate in the text that the yellow arrows point to LAMP1 structures at cell vertices (new Fig. 5d).

7. Figure 7e is missing the annotation +/- Rapalog.

This was fixed in the new version of Fig. 7e, now Fig. 8e.

8. Figure 6c Annotation of *GST-DLIC1* and *GST-DLIC-CT* is swapped.

This was fixed in the new version of Fig. 6c, now Fig. 7c.

9. Some of the clustering pictures in Fig4 are not convincing enough (compared to the quantification). For example, Δ RUN-GFP does not look as clustered as *RUFY3*-GFP, while the quantifications show similar average clustering.

We have replaced the image for Δ RUN by another image that is more representative of the quantitative data (new Fig. 5a). Both the images and the quantification suggest that this construct is a bit less effective at clustering endolysosomes, although the differences do not rise to the level of statistical significance.

Reviewer #3

The manuscript by Keren-Kaplan et al. is a well written study that describes identification of novel ARL8 GTPase effectors, RUN- and FYVE-domain containing proteins, RUFY3 and RUFY4 that are important for retrograde lysosome movement in cells. The authors use a combination of cell biology and biochemical methods to show that RUFY3 and RUFY4, but not other members of the RUFY family, associate with ARL8-lysosomes. The authors also map regions in RUFY3 important for an interaction with ARL8. Furthermore, the authors provide evidence that the retrograde movement of ARL8-lysosomes is mediated by RUFY3 and RUFY4 association with the dynein complex, although direct evidence of binding to dynein is only shown for RUFY3. Together with previous studies from this group and others showing that ARL8 can associate with kinesin adaptors to move lysosomes in an anterograde direction, this work provides important insights into how ARL8 regulates movement of lysosomes in cells.

The experiments are described in sufficient detail to be replicated and statistical analysis appears appropriate. I think that this study will be of high interest to the field of cellular trafficking, and I recommend its publication in the journal of Nature Communications.

We also thank this reviewer for the positive comments on our manuscript.

I have few minor suggestions described below:

1. In section titled: "ARL8B promotes recruitment of RUFY3 and RUFY4 to a juxtannuclear cluster of vesicles" authors conclude that ARL8 promotes the recruitment of RUFY3 and RUFY4 vesicles via the CC2 domain and that FYVE domain makes additional contributions to this recruitment. However, the contributions of RUFY4 domains are never tested. I would suggest clarifying this statement to focus only on domain contributions from RUFY3.

We thank the review for pointing out this misstatement. We have modified the text to refer only to domains of RUFY3.

2. The different RUFY3-GFP truncation constructs shown in inserts in figure 4e look very cytoplasmic, which is not fully consistent with images shown in figure 3d. Is there an increase in cytoplasmic localization of these constructs when co-expressed with LAMP1? If so, this would be unexpected and should be addressed in the text. It would also be helpful if the authors included full size images in the main figure or in supplementary figure.

The images in the insets of Fig. 4a look more cytosolic because 1) the constructs are overexpressed in order to maximize their effects on endolysosomes, and 2) the cells were fixed before staining, in contrast to those in Fig. 3d (now Fig. 3f), which were imaged live. We now clarify these differences in the text and figure legends.

3. There is a typo in the legend for figure 1, panel f is labeled with an e.

Fixed.

4. Figure 2 panel a, please add description of what is being blotted in the input.

We now indicate in the figure that the input is immunoblotted for the FLAG epitope.

5. Figure 2 panel e has RUN-GFP included although there is no description of this construct anywhere in

the text or figure legend. Similarly, there is RUN-FLAG included in panel f and g, but again there is no description of this construct. Please add description of these constructs.

We now describe in the text the results with the RUN-FLAG construct in reference to Fig. 2f, g.

6. In general, all labels for microscopy images are very small and hard to see. Could the authors please increase font size or rearrange labels to be above or below images, so it is easier to see them?

Whenever possible, we have increased the font size of the labels. In some cases, it was not possible for reasons of space.

7. Please add reference for the source of ARL8A-B-KO cells.

We added a reference for these cells (Keren-Kaplan and Bonifacino, 2021, *Curr. Biol.* **31**, 540-554 e545).

Reviewer #4 (Remarks to the Author)

In this manuscript, the authors identified RUFY3.1, a previously uncharacterized longer isoform of RUFY3, and RUFY4 as novel ARL8 effectors by using the MitoID method. They then found that both RUFY3.1 and RUFY4 promote retrograde transport of lysosomes through physical interaction with the dynein-dynactin motor complex and that their activity is required for accumulation of lysosomes in the juxtannuclear area of cells. Overall, the experiments are well conducted and there is good inclusion of positive and negative controls to reveal the selectivity and specificity of the observations. Thus, the study makes a significant contribution to the mechanistic understanding of ARL8-mediated lysosomal positioning that plays important roles in a wide range of cellular processes.

We thank this reviewer for the positive comments on our study.

Specific comments:

1. In Figure S1c, the authors showed that RUFY3.2, an originally described RUFY3 isoform, is present in the cytosol in HeLa cells, strongly suggesting that RUFY3.2 is not an ARL8 effector. In the present manuscript, however, it is not clear whether RUFY3.2 binds to ARL8. The authors should test their interaction biochemically.

We have now added mitochondrial relocation and pulldown analyses for RUFY3.1 vs RUFY3.2 (new Supplementary fig. 1c,d,g,h). The results show that the shorter RUFY3.2 isoform, lacking the FYVE domain and part of the CC2 domain, does not interact with ARL8.

2. Based on the results of Figure 3d, the authors concluded that in addition to the ARL8-interacting CC2 domain, the FYVE domain also contributes to the RUFY3.1 localization to cytoplasmic vesicles. Is RUFY3.1 localized to cytoplasmic vesicles via its FYVE domain independently of ARL8? The authors need to test the localization of RUFY3.1-deltaCC2 in ARL8A/B-KO cells.

We thank the reviewer for suggesting this very important experiment. In the new Supplementary fig. 2a, we show that ARL8 KO reduces overall vesicular staining for RUFY3-GFP and the juxtannuclear clustering of RUFY3-GFP vesicles, does not change the already dispersed distribution of the Δ CC2 construct, and abrogates the association of the Δ FYVE construct with vesicles. These findings confirm that the ARL8-binding CC2 domain is important for juxtannuclear clustering, while both the CC2 and the FYVE domain contribute to vesicle recruitment.

3. *In Figure 5i, expression of RUFY3-GFP or RUFY4-GFP promoted retrograde transport of lysosomes in axons but also "reduced the number of moving lysosomes". Why? Please discuss the possible reason. Does expression of RUFY3-GFP or RUFY4-GFP also affect lysosome transport in "dendrites"?*

In the new version of the manuscript, we explain more clearly that expression of RUFY3-GFP or RUFY4-GFP both reduces the total number of moving LAMP1 vesicles, and shifts the balance of the moving LAMP1 vesicles towards retrograde transport. This is consistent with RUFY3-GFP or RUFY4-GFP promoting retrograde over anterograde transport. As in non-neuronal cells, the reduction in LAMP1 tracks and LAMTOR4 puncta in the axon (new Fig. 6g and Supplementary Fig. 4a,b; see response to Reviewer #1, point d) likely results from accumulation of endolysosomes in the juxtannuclear area of the neurons. We now make this explanation clearer in the text.

RUFY3-FLAG and RUFY4-FLAG also colocalize with LAMP1-GFP and endogenous LAMTOR4 in dendrites (Fig. 6a-d). However, because dendrites have mixed microtubule polarity, it is harder to assess whether RUFY3-FLAG and RUFY4-FLAG alter the distribution or movement of lysosomes. For this reason, we limited our imaging of moving endolysosomes to axons, where microtubules are uniformly polarized and it is easier to assess anterograde vs. retrograde movement.

4. *Distinct Rab binding activities of the RUFY family members in the Discussion section is important (lines 397-404), but the information about the Rab33A binding activity of RUFY3.2 is missing.*

In the new version of the Discussion, we mention the fact that RUFY3.2 interacts with RAB33, and point out that the significance of this interaction for recruitment of RUFY3.2 to endolysosomes remains to be addressed.

5. *Typos. (line 40) unles; (line 357) RUFY3.3; (line 422) Once possibility; (line 683) Fig. 3c,4b should read Fig. 3b, 4b.*

Figure 6c seems to be mis-labeled. The far-right lane should be the GST-DLIC1 full-length protein.

Page numbers of several references are missing (e.g., lines 973,1026, 1064, 1082, 1096, and 1114).

All of these typos were fixed.

Reviewer #5:

Keren-Kaplan et al. use an alternative BioID2 assay called MitoID to identify proximal proteins of ARL8A and ARL8B. This resulted in the identification of RUFY3 and RUFY4 proteins that bind to the GTP-bound form of ARL8. The authors further demonstrate the interaction with dynein dynactin and link juxtannuclear redistribution of lysozymes under various conditions with the complex that they have described in this study.

For the proteomics part, the design of the experiment is not very clear in the current manuscript. While the authors briefly mention in the methods section that 3 biological replicates were used (with 3 '.raw' in the data files), this is not clear from the results part, nor from fig. S1A or the processed tables. It would be better to make this clear to the readers in the figures and the text. Accordingly, the data processing to obtain Figures 1B and 1C and Supplementary dataset 1 should be better explained in the manuscript. BioID and BioID2 data tend to generate a lot of background (especially upon massive overexpression as is the case here) so careful data analysis and processing is essential and should be transparent. For example, how exactly was the abundance ratio obtained and how was this transformed to fit the 0-100 axis?

We now provide a more detailed description of the MitoID and mass spectrometry procedures in the legend to Fig. 1 and the Methods sections. This includes a statement in both the Fig. 1 legend and Methods that 3 biological replicates were used.

MitoID was chosen precisely because the targeting of the bait constructs to mitochondria provides for a more uniform background of non-specific hits, irrespective of the original localization of the baits. This is now mentioned in the first paragraph of the Results section.

The default value of 100 was set as the maximum fold change allowed. For example, if the calculated ratios are 50, 80, 120 and 150 for proteins A, B, C, and D, the ratios reported by PD software are 50, 80, 100, 100 for proteins A, B, C and D. The abundance ratio was not transformed in any way. It was plotted on the graph as presented in the Excel file.

Moreover, the proteomics data should be made available in a public repository, so it becomes available to the community. It is not clear whether this is planned.

Proteomics raw data and search results were deposited in the MassIVE repository and will be available upon publication: Title: Tal Keren-Kaplan and Juan Bonifacino LFQ data, link: <ftp://massive.ucds.edu/MSV000087741/>.

An Excel spreadsheet with all the proteins identified in the study is also included as Supplementary Data 1.

Along these lines, it is also worthwhile to elaborate briefly on the benefits of MitoID over classical BioID (or BioID2) to better motivate the use of this interesting method to the readers.

In the Results section (under section "Identification of RUFY3 and RUFY4 as ARL8 effectors"), we now mention that one advantage of MitoID over BioID is a more uniform identification of non-specific proteins due to the targeting of all constructs to mitochondria, irrespective of what compartment they normally associate with in the absence of the mitochondrial-targeting signal (e.g., endolysosomes vs. cytosol).

Fig 4g. shows RT-qPCR analysis upon knockdown of the RUFY3. While I appreciate the other data supporting the link between RUFY3 and lysosomal localization, perturbation is important. The authors

should clarify if this was a single siRNA or a pool of siRNAs. In the case of a single siRNA, it would be best to add another siRNA to eliminate the possibility of off-target effects. In the case of a pool, the pool should be deconvoluted to find (hopefully) at least 2 decent siRNAs that give the same phenotype.

We thank the reviewer for this comment. We now indicate in the legend to the new Fig. 5 that RUFY3 was knocked down using a SMARTpool. In addition, in the new Supplementary Fig. 3, we show RUFY3 KD and LAMP1 redistribution data for single siRNAs from the SMARTpool. Three of the 4 siRNAs are effective in knocking down RUFY3 and causing endolysosome dispersal.

Minor comments

1. Line 110. The originally described MitoID procedure attaches BioID and the mitochondrial targeting sequences C-terminally from the Ras family protein under investigation. Please provide some more explanation for switching the modules in this study.

The RAB GTPases used in the original study are anchored to membranes via their C-terminus. In contrast, ARL GTPases, including the ARL8 used in our study, are anchored to membranes via their N-terminus, thus the different design of our constructs. We now briefly explain this in the first paragraph of the results and the corresponding section of the Methods (Recombinant DNAs).

2. Line 112. Provide a reference for the ARL8 mutants that lock the protein in the GDP and GTP-bound state.

We added a reference for these mutations.

3. Line 601: I assume that TECP is referring to the reducing chemical tris (2-carboxyethyl)phosphine so this should be TCEP.

We corrected the abbreviation to TCEP and explained its meaning (Tris(2-carboxyethyl)phosphine).

4. Methods line 710-712. Transfection method and incubation time after transfection are not mentioned.

The transfection protocol was described in the “Cell culture and treatments” section.

5. Methods line 718. “Following incubation, cells were washed” à I think this should be “beads were washed”?

Corrected.

6. Fig 2b. On the bottom panel, many aspecific signals can be seen with very different intensities, please indicate the band sizes expected for each FLAG-tagged construct (for example in the figure legend)

We added asterisks indicating the undegraded proteins and included the expected molecular masses in the figure legend.

7. Line 208, Fig 3c makes use of HeLa ARL8A-B KO cells, but these cells are not mentioned anywhere in the methods. Please describe their origin or how you established them and provide evidence of successful KO if these cells have not been published elsewhere.

The HeLa ARL8A-B KO cells were described previously (Keren-Kaplan and Bonifacino, 2021, *Curr. Biol.* **31**, 540-554 e545). We have added the reference to the text.

8. Fig 7e. Indicate which bars refer to rapalog-treated cells. Also check RUFY4 condition (NT and DHC siRNA)

We modified Fig. 7e (now Fig. 8e) to indicate the rapalog-treated cells. We also corrected the labeling error for RUFY4.

REVIEWERS' COMMENTS

Reviewer #1 (Remarks to the Author):

The study by Keren-Kaplan et al identifying 2 novel effectors of the small GTPase Arl8, namely RUFY3 and RUFY4 is interesting and novel in that it identifies these new effectors of ARL8 GTPase and links this GTPase to retrograde transport for the first time. As mentioned before in initial review the topic is important and the study itself is very well done with well-designed experiments, striking images and quantitative data. The authors have addressed the minor issues raised in initial review with additional experiments, quantitative data they have included and/or clarifications in text. I recommend publication of this interesting study.

Reviewer #2 (Remarks to the Author):

The authors addressed my points

Reviewer #3 (Remarks to the Author):

The authors have addressed all my comments.

Reviewer #4 (Remarks to the Author):

In the revised manuscript, the authors properly addressed all of the concerns raised by this reviewer. Thus, I would like to recommend the revised manuscript for publication in its present form.

Reviewer #5 (Remarks to the Author):

Thank you for introducing clarifications in the manuscript. Please rephrase 'using four pooled siRNAs' to 'using a pool of four siRNAs'.

Manuscript has further improved and can be accepted now.

Responses to Reviewers

Authors' responses are indicated in red.

Reviewer #1 (Remarks to the Author):

The study by Keren-Kaplan et al identifying 2 novel effectors of the small GTPase Arl8, namely RUFY3 and RUFY4 is interesting and novel in that it identifies these new effectors of ARL8 GTPase and links this GTPase to retrograde transport for the first time. As mentioned before in initial review the topic is important and the study itself is very well done with well-designed experiments, striking images and quantitative data. The authors have addressed the minor issues raised in initial review with additional experiments, quantitative data they have included and/or clarifications in text. I recommend publication of this interesting study.

Reviewer #2 (Remarks to the Author):

The authors addressed my points

Reviewer #3 (Remarks to the Author):

The authors have addressed all my comments.

Reviewer #4 (Remarks to the Author):

In the revised manuscript, the authors properly addressed all of the concerns raised by this reviewer. Thus, I would like to recommend the revised manuscript for publication in its present form.

We are pleased to have been able to address all the comments by Reviewers #1-4.

Reviewer #5 (Remarks to the Author):

Thank you for introducing clarifications in the manuscript. Please rephrase 'using four pooled siRNAs' to 'using a pool of four siRNAs'.

Manuscript has further improved and can be accepted now.

The text was changed as requested by this reviewer. We thank the review for recommending acceptance.